# m⁶A modification of U6 snRNA modulates usage of two major classes of pre-mRNA 5' splice site

Matthew T Parker[1], Beth K Soanes[2], Jelena Kusakina[2], Antoine Larrieu[2], Katarzyna Knop[1†], Nisha Joy[1], Friedrich Breidenbach[1,3], Anna V Sherwood[1‡], Geoffrey J Barton[1], Sebastian M Fica[4], Brendan H Davies[2]*, Gordon G Simpson[1,5]*

[1]School of Life Sciences, University of Dundee, Dundee, United Kingdom; [2]Centre for Plant Sciences, School of Biology, Faculty of Biological Sciences, University of Leeds, Leeds, United Kingdom; [3]RNA Biology and Molecular Physiology, Faculty of Biology, Bielefeld University, Bielefeld, Germany; [4]Department of Biochemistry, University of Oxford, Oxford, United Kingdom; [5]Cell & Molecular Sciences, James Hutton Institute, Invergowrie, United Kingdom

**\*For correspondence:**
b.h.davies@leeds.ac.uk (BHD);
g.g.simpson@dundee.ac.uk
(GGS)

**Present address:** †Cancer Research UK Beatson Institute, Glasgow, United Kingdom; ‡Section for Computational and RNA Biology, Department of Biology, University of Copenhagen, Copenhagen, Denmark

**Competing interest:** The authors declare that no competing interests exist.

## Abstract

Alternative splicing of messenger RNAs is associated with the evolution of developmentally complex eukaryotes. Splicing is mediated by the spliceosome, and docking of the pre-mRNA 5' splice site into the spliceosome active site depends upon pairing with the conserved ACAGA sequence of U6 snRNA. In some species, including humans, the central adenosine of the AC<u>A</u>GA box is modified by $N^6$ methylation, but the role of this m⁶A modification is poorly understood. Here, we show that m⁶A modified U6 snRNA determines the accuracy and efficiency of splicing. We reveal that the conserved methyltransferase, FIONA1, is required for *Arabidopsis* U6 snRNA m⁶A modification. *Arabidopsis fio1* mutants show disrupted patterns of splicing that can be explained by the sequence composition of 5' splice sites and cooperative roles for U5 and U6 snRNA in splice site selection. U6 snRNA m⁶A influences 3' splice site usage. We generalise these findings to reveal two major classes of 5' splice site in diverse eukaryotes, which display anti-correlated interaction potential with U5 snRNA loop 1 and the U6 snRNA AC<u>A</u>GA box. We conclude that U6 snRNA m⁶A modification contributes to the selection of degenerate 5' splice sites crucial to alternative splicing.

## Editor's evaluation

This is an important paper reporting that an adenosine methyltransferase in the model plant Arabidopsis functions to target a key RNA component of the spliceosome, as in fission yeast, and thereby contributes to intron recognition as well as flowering timing. By contrast, the authors report no major role for the methyltransferase in targeting mRNAs, as reported in several previous studies in Arabidopsis. Overall, the approaches are convincing, although a correlative analysis identifying an intronic sequence feature characteristic of methyltransferase-sensitive introns is not followed up with tests to establish causality of this feature.

## Introduction

Split genes are a defining characteristic of eukaryotic genomes (*Plaschka et al., 2019*). During pre-mRNA transcription, intervening sequences (introns) are excised, and the flanking sequences (exons) are spliced together. In developmentally complex eukaryotes, such as humans or *Arabidopsis*, most

**eLife digest** All the information necessary to build the proteins that perform the biological processes required for life is encoded in the DNA of an organism. Making these proteins requires the DNA sequence of a gene to be transcribed into a 'messenger RNA' (mRNA), which is then processed into a final, mature form. This blueprint is then translated to assemble the corresponding protein.

When an mRNA is processed, segments of the sequence that do not code for protein are removed and the remaining coding sequences are joined together in the right order. An intricate molecular machine known as the spliceosome controls this mechanism by recognising the 'splice sites' where coding and non-coding sequences meet. Depending on external conditions, the spliceosome can 'pick-and-mix' the coding sequences to create different processed mRNAs (and therefore proteins) from a single gene. This alternative splicing mechanism is often used to regulate when certain biological processes take place based on environmental cues; for example, the splicing of genes which control the timing of plant flowering is sensitive to ambient temperatures.

To investigate this mechanism, Parker et al. focused on *Arabidopsis thaliana*, a plant that blooms later when temperatures are low. This precise timing partly relies on a gene whose mRNA is efficiently spliced in the cold, resulting in an active form of its protein that blocks blooming. Parker et al. grew and screened many *A. thaliana* plants to find individuals that could flower early in the cold, in which splicing of this gene was disrupted.

A mutant fitting these criteria was identified and subjected to further investigation, which revealed that it could not produce FIONA1. In non-mutant plants, this enzyme chemically modifies one of the components of the spliceosome, a small nuclear RNA known as U6. Parker et al found that there are two types of splice site – one more likely to interact with U6 and another that preferentially interacts with another small nuclear RNA, U5. When FIONA1 is inactive (such as in the mutant identified by Parker et al.), splice sites that tend to strongly interact with U5 are selected. However, when the enzyme is active, splice sites that tend to bind with the chemically modified U6 are used instead.

Further work by Parker et al. showed that these two types of splice sites ('preferring' either U5 or U6) are found in equal proportions in the genomes of many species, including humans. This suggests that Parker et al. have uncovered an essential feature of how genomes are organised and splicing is controlled.

protein-coding genes have introns, and the *cis*-elements controlling splicing are degenerate. Such sequence variation facilitates alternative splice site selection that, in turn, permits the regulation of mRNA expression and the production of functionally different protein isoforms (Lee and Rio, 2015; *Nilsen and Graveley, 2010*). Consistent with this, alternative splicing is the foremost genomic predictor of developmental complexity (*Chen et al., 2014*).

Pre-mRNA splicing is carried out by the spliceosome (*Plaschka et al., 2019*; *Wilkinson et al., 2020*). This large, dynamic molecular machine comprises more than 100 proteins and 5 uridylate-rich snRNAs (UsnRNAs). Splicing requires two sequential transesterification reactions. The first reaction, called branching, occurs when the 2' hydroxyl of the conserved intron branchpoint adenosine performs a nucleophilic attack on the 5' splice site (5'SS), resulting in a cleaved 5' exon and a branched lariat-intron intermediate. The second reaction, exon ligation, occurs via nucleophilic attack of the 3' hydroxyl of the 5' exon on the 3' splice site (3'SS). Typically, the 5'SS is first recognised by U1 snRNP, whilst U2 snRNP recognises the branchpoint sequence of pre-mRNA. The preassembled U4/U6.U5 tri-snRNP subsequently joins the spliceosome. The conserved U6 snRNA ACAGA sequence replaces U1 snRNP at the 5'SS and loop 1 of U5 snRNA binds 5' exon sequences adjacent to the 5'SS. In humans, selection of the 5'SS occurs during this hand-off to U6 and U5 snRNAs, and this step is decoupled from formation of the active site, thus potentially allowing for plasticity of 5'SS selection (*Charenton et al., 2019*; *Fica, 2020*). Subsequently, U6 pairs with U2 snRNA and intramolecular U6 snRNA interactions facilitate the positioning of the catalytic metal ions to effect branching and exon ligation (*Wilkinson et al., 2020*).

U6 snRNA is the most highly conserved UsnRNA, reflecting its crucial roles at the active site of the spliceosome (*Wilkinson et al., 2020*). Different base modifications are found in each of the UsnRNAs, including U6 snRNA, but the role of these modifications is poorly understood (*Morais et al., 2021*).

The essential ACAGA sequence of *Saccharomyces cerevisiae* U6 snRNA is unmodified. However, in other species including *Schizosaccharomyces pombe* (*Gu et al., 1996*), the plant *Vicia faba* (*Kiss et al., 1987*), and human (*Shimba et al., 1995*), the central adenosine in the corresponding sequence is modified by methylation at the N6 position: $AC^{m6}\underline{A}GA$. In *S. cerevisiae*, U6 snRNA ACAGA recognises the stringently conserved GUAUGU sequence at the 5′ end of introns, with the central adenosine making a Watson-Crick base pair with the almost invariant U at the +4 position of the 5′SS (5′SS $U_{+4}$) (*Neuvéglise et al., 2011*; *Wan et al., 2019*). Conversely, in species with m6A-modified U6 snRNA, the 5′SS can be degenerate and the identity of the base at the +4 position varies but is usually enriched for A. Therefore, the pairing between the 5′SS and the U6 snRNA ACAGA box is unlikely to be driven primarily by canonical Watson-Crick base pairs in these species. The precise function of the U6 snRNA m6A modification is unknown.

In humans, the U6 snRNA AC**A**GA sequence is methylated by the conserved methyltransferase METTL16 (*Aoyama et al., 2020*; *Pendleton et al., 2017*; *Warda et al., 2017*). This modification depends upon a specific sequence and distinct structural features of U6 snRNA that are recognised by METTL16. Hairpin sequences that mimic these features of U6 snRNA are found within the 3′UTR of S-adenosylmethionine (SAM) synthetase *MAT2A* mRNA and are also methylated by METTL16 (*Pendleton et al., 2017*; *Shima et al., 2017*). SAM synthetase is the enzyme responsible for production of the methyl donor SAM, which is required for methylation reactions in the cell. Binding of METTL16 to *MAT2A* mRNA influences splicing and/or stability, regulating MAT2A expression and SAM levels (*Pendleton et al., 2017*; *Shima et al., 2017*). In *Caenorhabditis elegans*, the METTL16 orthologue, METT-10, methylates U6 snRNA and also influences SAM levels by targeting SAM synthetase (*sams*) genes (*Mendel et al., 2021*). METT-10 methylates the 3′SS of *sams-3/4* intron 2, although in a different sequence and structural context to U6 snRNA (*Mendel et al., 2021*). In *S. pombe*, the METTL16 orthologue, MTL16, appears to target only U6 snRNA (*Ishigami et al., 2021*). *S. pombe* mutants defective in MTL16 function show less efficient splicing of some introns, resulting in increased levels of intron retention. The *S. pombe* introns sensitive to loss of MTL16 function are distinguished by having adenosine at the +4 position of the intron (5′SS $A_{+4}$). However, the effect of 5′SS $A_{+4}$ is partially suppressed in introns that have stronger predicted base pairing between U5 snRNA loop 1 and the 5′ exon (*Ishigami et al., 2021*).

Alternative splicing plays crucial roles in gene regulation, affecting diverse biological processes including the control of flowering in response to ambient temperature (*Airoldi et al., 2015*; *Capovilla et al., 2017*). Slight changes in ambient temperature can profoundly influence flowering time (*Andrés and Coupland, 2012*). Indeed, documented changes in the flowering times of many plant species have provided some of the best biological evidence of recent climate change (*Fitter and Fitter, 2002*). At cooler ambient temperatures, pre-mRNAs encoding repressors of *Arabidopsis* flowering, FLM (FLOWERING LOCUS M) and MAF2 (MADS AFFECTING FLOWERING 2), are efficiently spliced (*Airoldi et al., 2015*; *Balasubramanian et al., 2006*; *Lee et al., 2013*; *Lutz et al., 2015*; *Posé et al., 2013*; *Scortecci et al., 2001*). As a result, FLM and MAF2 proteins are produced at cooler temperatures and form higher-order protein complexes with other MADS box factors, such as MAF3 (*MADS AFFECTING FLOWERING 3*), SVP (*SHORT VEGETATIVE PHASE*), and FLC (*FLOWERING LOCUS C*), to directly repress the expression of the genes *FT (FLOWERING LOCUS T)* and *SOC1 (SUPPRESSOR OF OVEREXPRESSION OF CO 1)*, which promote flowering (*Gu et al., 2013*). At elevated ambient temperatures, the splicing of *MAF2* and *FLM* introns is adjusted, in different ways, effecting increased levels of non-productive transcripts (*Airoldi et al., 2015*; *Balasubramanian et al., 2006*; *Lee et al., 2013*; *Lutz et al., 2015*; *Posé et al., 2013*; *Scortecci et al., 2001*). Consequently, the abundance of active MAF2 and FLM proteins is compromised at elevated temperatures; repression of genes promoting flowering is relieved, and flowering is enabled (*Airoldi et al., 2015*; *Capovilla et al., 2017*; *Posé et al., 2013*; *Rosloski et al., 2013*). In the case of *MAF2*, this temperature-responsive splicing is characterised by a progressive increase in retention of intron 3 as ambient temperature increases (*Airoldi et al., 2015*). The thermosensory mechanism that underpins the temperature-responsive alternative splicing of *MAF2* and *FLM* is unknown.

We conducted a two-step mutant screen for factors required for the efficient splicing of *MAF2* pre-mRNA intron 3 at cooler temperatures. This screen identified an early flowering mutant disrupting the gene encoding the *Arabidopsis* METTL16 orthologue, FIONA1 (FIO1; *Kim et al., 2008*). We show that FIO1 is required for m6A modification of U6 snRNA. We detect widespread changes in pre-mRNA

splicing in *fio1* mutants that can be explained by the identity of the base at the 5'SS +4 position of affected introns and by cooperative roles for U5 and U6 snRNA in 5'SS selection. We also reveal that U6 snRNA interactions at 5'SSs affect 3'SS choice. Analysis of other annotated genomes reveal the existence of two major classes of 5'SS. Our findings suggest cooperative and compensatory roles of U5 and U6 snRNA may contribute to 5'SS selection in a range of eukaryotes.

## Results

### Identification of *FIO1* in a two-step screen for early flowering mutants with increased retention of *MAF2* intron 3

To identify factors responsible for promoting efficient splicing of *MAF2* intron 3 at low ambient temperature, we performed a two-step genetic screen using ethyl methanesulfonate (EMS) as a mutagen (*Figure 1A*). In the first step, we screened 15,000 EMS lines for early flowering *Arabidopsis* mutants at 16°C. In the second step, the earliest flowering 100 individuals were re-screened for enhanced *MAF2* intron 3 retention at 16°C, using RT-PCR. Two independent mutants (EMS-129 and EMS-213) showed early flowering and increased levels of *MAF2* intron 3 retention at 16°C. We focus here on EMS-129 (*Figure 1—figure supplement 1A*). The causal mutation for the early flowering phenotype of EMS-129 was mapped by back-crossing to the M0 parental line to generate a segregating population (*Figure 1—figure supplement 1B*). Early flowering plants and aphenotypic sisters were grouped into separate pools for genomic DNA sequencing. EMS-induced G to A transitions were identified, and allele fractions were compared between the two pools (*Javorka et al., 2019*). A total of 19 homozygous G to A transitions in a 2.6 Mb region on chromosome 2 were associated with the early flowering phenotype (*Figure 1B*). Of these, only the SNP at position 9,041,454 of chromosome 2, which disrupts the 5'SS of AT2G21070 intron 2, was predicted to have a major impact on gene expression. An EMS mutation in AT2G21070 has previously been described and named *fio1-1* (*Kim et al., 2008*). Crosses between EMS-129 and *fio1-1* confirmed allelism because 100% of F1 progeny from EMS-129 lines crossed to *fio1-1* flowered early (35 plants) compared to 0% of F1 progeny from EMS-129 lines crossed to Col-0 (15 plants). We isolated an independent transfer-DNA insertion in AT2G21070 (*Alonso et al., 2003*), which we refer to as *fio1-3*. Henceforth, we refer to EMS-129 as *fio1-4*. We summarise the *fio1-1*, *fio1-3*, and *fio1-4* alleles in *Figure 1C*. *fio1-1* and *fio1-3* show increased levels of *MAF2* intron 3 retention at low temperature (*Figure 1D*) and like *maf2* mutants, flower early (*Figure 1E–F*). *fio1* alleles share additional visible phenotypes, including reduced apical dominance and stature (*Figure 1F*). *FIO1* encodes the *Arabidopsis* orthologue of the human methyltransferase, METTL16 (*Kim et al., 2008*; *Pendleton et al., 2017*).

### FIO1-dependent methylation of poly(A)+ RNA is rare or absent

To reveal which RNAs were methylated by FIO1, we first used two orthogonal approaches to determine the impact of FIO1 on poly(A)+mRNA m⁶A modification: liquid chromatography tandem mass spectrometry (LC-MS/MS) and nanopore direct RNA sequencing (DRS). As a positive control for these experiments, we included a hypomorphic *fip37-4* allele defective in the *Arabidopsis* orthologue of the METTL3 m⁶A writer complex component WTAP (*Zhong et al., 2008*).

LC-MS/MS analysis (*Figure 2A*) revealed an 81.8% decrease in m⁶A levels in *fip37-4* compared to Col-0 (*t* test p=$2.2 \times 10^{-6}$). In contrast, a 6.7% decrease in m⁶A levels was observed in poly(A)+RNA purified from *fio1-1* (*t* test p=0.40). A similarly modest decrease in mRNA m⁶A levels has recently been reported for other *fio1* alleles (*Wang et al., 2022*).

We previously used nanopore DRS to map m⁶A sites dependent on the *Arabidopsis* METTL3-like writer complex component VIRILIZER (VIR), revealing m⁶A enriched in a DR**m⁶A**CH consensus in the 3' terminal exon of protein coding mRNAs (*Parker et al., 2020*). We used a similar nanopore DRS experiment to map FIO1-dependent m⁶A. We sequenced poly(A)+RNA purified from four biological replicates each of wild-type (Col-0), *fio1-1* and *fip37-4*, resulting in a total of 22.7 million mapped reads. The corresponding sequencing statistics are detailed in *Supplementary file 1*. We used the software tool Yanocomp to map differences in mRNA modifications (*Parker et al., 2021a*). Applying this approach, we identified 37,861 positions that had significantly different modification rates in a three-way comparison between Col-0, *fip37-4*, and *fio1-1* (*Figure 2B*). Of these, 97.9% had significant changes in modification rate in a pairwise comparison between *fip37-4* and Col-0. In contrast, only

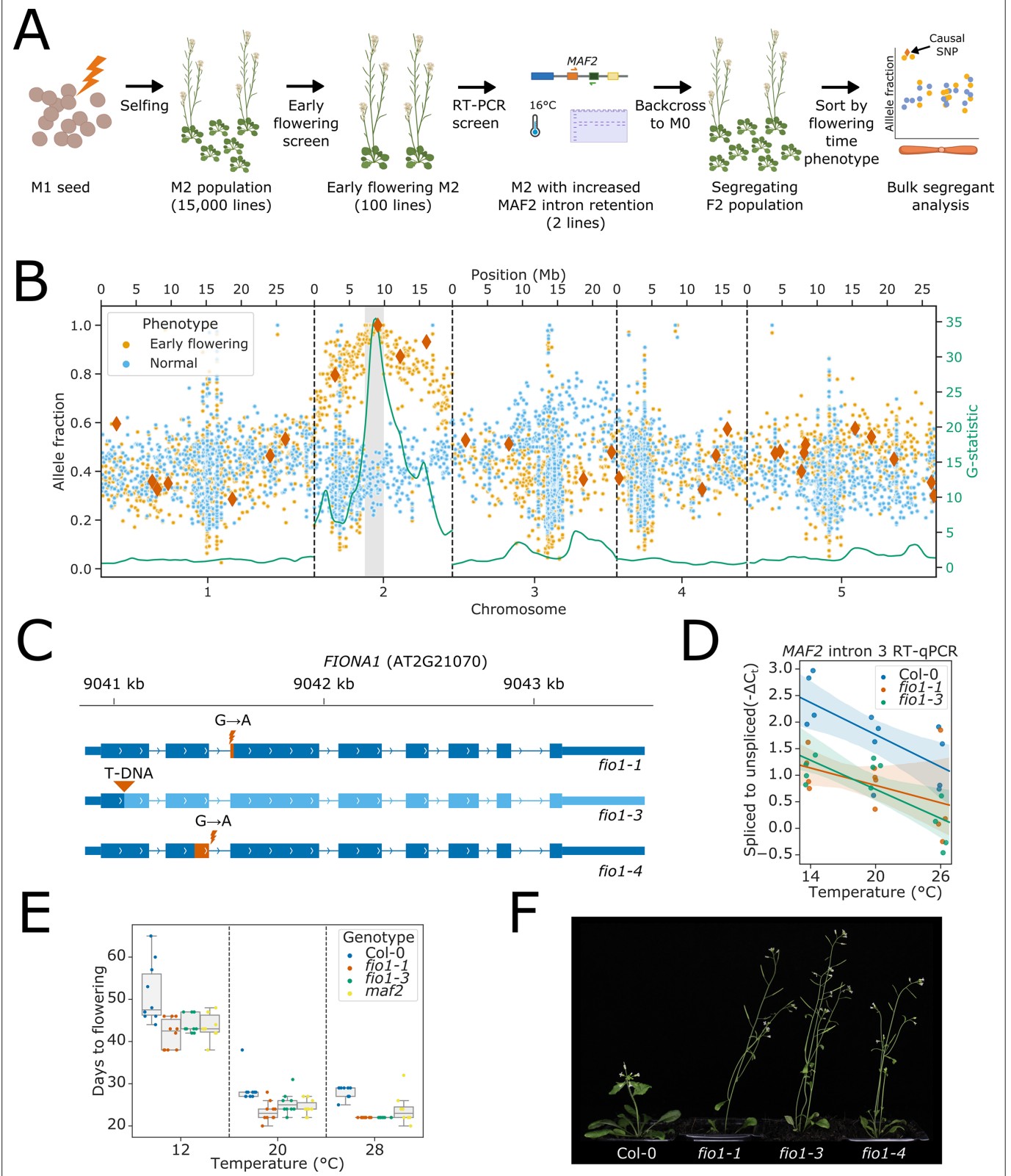

**Figure 1.** Loss of *FIO1* causes early flowering and reduced splicing of pre-mRNA encoding the floral repressor MAF2. (**A**) Schematic showing the format of the two-step mutant screen. (**B**) Scatter plot showing allele fractions of ethyl methanesulfonate (EMS)-induced G to A transitions in pooled phenotypically normal plants (blue) and early flowering sisters (orange). Dark orange diamonds show SNPs predicted to have a significant impact on the functional expression of a protein-coding gene (nonsense mutations and splice site mutations). The green line shows the test statistic for the G-test

*Figure 1 continued on next page*

*Figure 1 continued*

between the allele fractions found in early flowering and normal plants, which has been smoothed using a tri-cube kernel with a window size of two megabases (Mb). The 2.6 Mb mapping interval containing *FIO1* is highlighted in grey. (**C**) Gene track showing the three *fio1* mutant alleles used in this study. *fio1-1* is an EMS mutant with a G→A transition at the –1 position of the 3′ splice site (3′SS) in intron 2 of *FIO1* (**Kim et al., 2008**). This causes activation of a cryptic 3′SS 15 nt downstream, and the loss of 5 aa of sequence from the *FIO1* open reading frame (shown in orange). *fio1-3* is a T-DNA insertion mutant (SALK_084201) in the first exon of the *FIO1*, disrupting the gene (region downstream of insertion shown in light blue). *fio1-4* is an EMS mutant with a G→A transition at the +1 position of the 5′ splice site (5′SS) in intron 2 of *FIO1*. This causes activation of a cryptic 5′SS 69 nt upstream, and the loss of 23 aa of sequence from the *FIO1* open reading frame (shown in orange). (**D**) Changes in splicing efficiency determined by Reverse transcription-quantitative PCR (RT-qPCR) visualised as a regression scatterplot showing the change in spliced to retained ratio of *MAF2* intron 3 in *fio1-1* and *fio1-3* at a range of temperatures. Shaded regions show bootstrapped 95% CI for regression lines. (**E**) Boxplot showing the change in flowering time (days to flowering) observed in the *fio1-1*, *fio1-3*, and *maf2* mutants at a range of temperatures. (**F**) Photographs showing the early flowering phenotypes of *fio1-1*, *fio1-3*, and *fio1-4* mutants.

The online version of this article includes the following source data and figure supplement(s) for figure 1:

**Source data 1.** Ethyl methanesulfonate (EMS) mutations identified in early flowering and *MAF2* splicing screen.

**Source data 2.** Sanger sequencing products for *FIO1* cDNAs in *fio1-1* and *fio1-4* mutant.

**Source data 3.** Sanger sequencing product alignments for *FIO1* cDNAs in *fio1-1* and *fio1-4* mutant.

**Source data 4.** MAF2 intron 3 splicing quantitative PCR (qPCR) data for Col-0, *fio1-1I* and *fio1-3* mutants.

**Source data 5.** Flowering time data for *fio1-1*, *fio1-3*, and *maf2* mutants.

**Source data 6.** RT-PCR screening of *MAF2* intron 3 splicing in early flowering mutants.

**Figure supplement 1.** (**A**) RT-PCR products separated by agarose gel electrophoresis used to identify enhanced *MAF2* intron 3 retention in the *EMS-129* line, later renamed to *fio1-4*. (**B**) Photograph of the F2 segregating population used to map *fio1-4*.

7.6% had altered modification rates in *fio1-1*. Of these, 85.8% also had altered modification rates in *fip37-4* (**Figure 2B**), with larger effect sizes (**Figure 2—figure supplement 1A**), indicating that they are METTL3-like writer complex-dependent m6A sites whose modification rate is indirectly affected by loss of FIO1. The FIP37-dependent modification sites, including those with small effect-size changes in *fio1-1*, were found in a DRACH consensus in 3′UTR regions (**Figure 2—figure supplement 1B–C**), consistent with established features of m6A modifications deposited by the *Arabidopsis* METTL3-like writer complex (**Parker et al., 2020**).

We used the antibody-based technique miCLIP to perform orthogonal validation of predicted modification sites (**Parker et al., 2020**): *FIP37*-dependent m6A sites were well supported, with 52.8% less than 5 nt from an miCLIP peak (**Figure 2B**). In contrast, of the 408 sites discovered only in *fio1-1*, just 21.3% (87) were less than 5 nt from an miCLIP peak, indicating that the majority are likely to be false positives. The 87 positions with miCLIP support were found in a DRACH consensus in 3′UTRs, suggesting that they are false negative *FIP37*-dependent m6A sites rather than genuine FIO1-dependent sites (**Figure 2—figure supplement 1D–E**).

We did not find *FIO1*-dependent methylation changes in transcripts of the four *Arabidopsis* homologues of SAM-synthetase (**Figure 2—figure supplement 2A–D**). Consequently, the METTL16-dependent regulation of SAM homeostasis identified in metazoans (**Mendel et al., 2021**; **Pendleton et al., 2017**; **Shima et al., 2017**) appears not to be conserved in *Arabidopsis*. Consistent with previous evidence that m6A in 3′UTRs affects cleavage and polyadenylation (**Parker et al., 2020**), we identified 1104 genes with altered poly(A) site choice in *fip37-4*, compared to only 49 in *fio1-1* (**Figure 2—figure supplement 1F**). Overall, these findings indicate that FIO1-dependent m6A sites in poly(A)+mRNA are rare or absent.

## m6A modification of U6 snRNA depends upon FIO1

Since FIO1 orthologues methylate U6 snRNA, we asked if FIO1 was required to modify *Arabidopsis* U6 snRNA. We performed two experiments using two independent anti-m6A antibodies to immuno-purify methylated RNAs from Col-0 and *fio1* alleles. For control, we used U2 snRNA, which is m6A methylated in humans, but not by METTL16 (**Chen et al., 2020**). Using RT-qPCR, we could detect equivalent levels of U6 snRNA in the input RNA purified from different genotypes, suggesting that the abundance of U6 snRNA is unaffected by loss of FIO1 function (**Figure 2—figure supplement 3A–B**). We detected enrichment of U6 and U2 snRNAs in RNA immunopurified with anti-m6A antibodies from Col-0 (**Figure 2C–D**). The enrichment of U2 snRNA in these experiments was unaffected by loss of

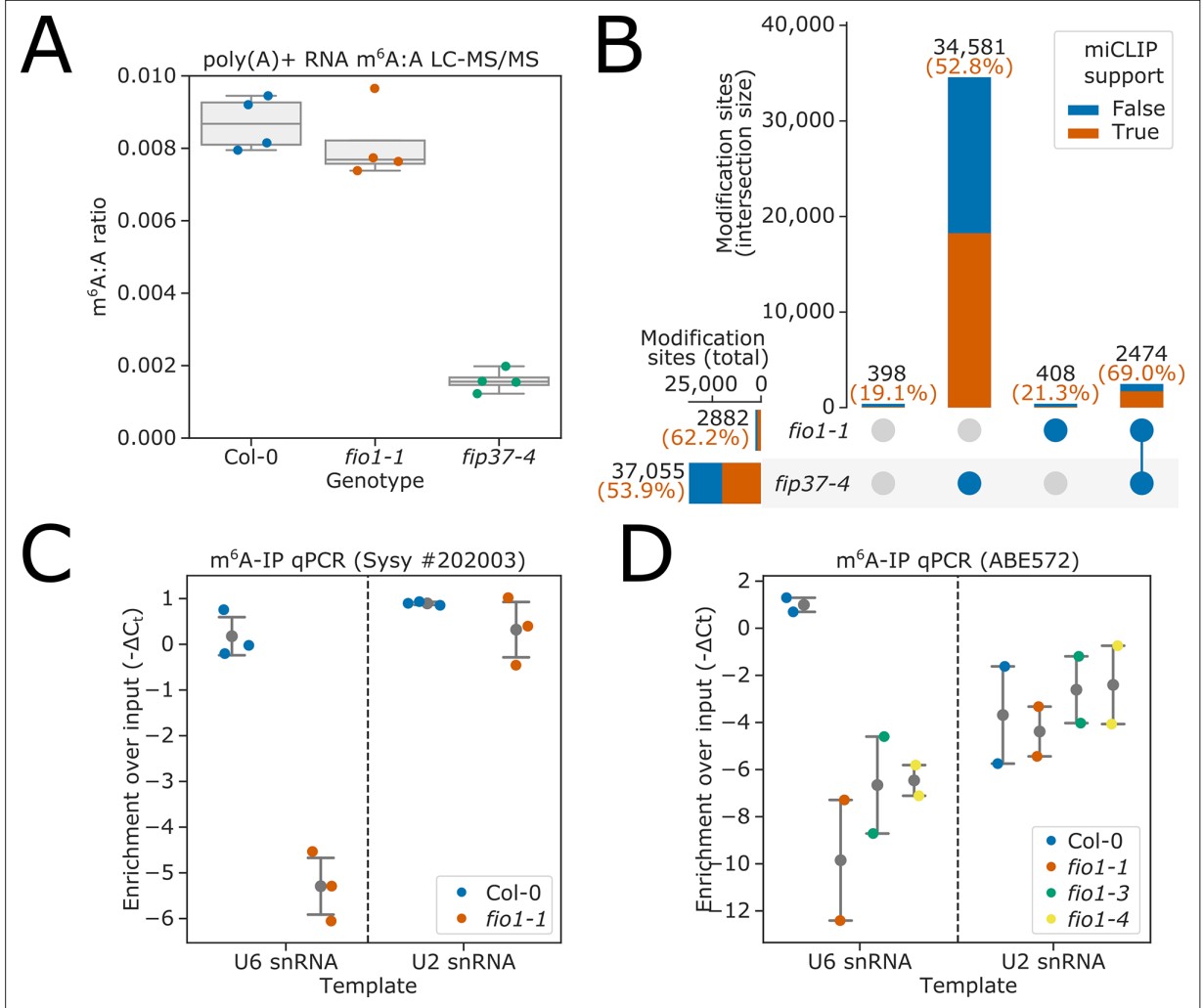

**Figure 2.** FIO1-dependent m⁶A modification of poly(A)+mRNA is rare or absent, but FIO1 is required for U6 snRNA methylation. (**A**) Liquid chromatography tandem mass spectrometry (LC-MS/MS) analysis. Boxplots showing that the ratio of m⁶A to A in poly(A)+mRNA is only modestly reduced in *fio1-1* mutants compared to Col-0. In contrast, m⁶A levels are significantly reduced in the *fip37-4* mutant. (**B**) The intersection of m⁶A modification sites detected by nanopore direct RNA sequencing of poly(A)+RNA purified from Col-0, *fip37-4*, and *fio1-1*, visualised using an upset plot. All sites shown have significant differences in modification level in a three-way comparison between *fip37-4*, *fio1-1*, and Col-0. Bars show size of intersections between sites which are significant in each two-way comparison. Total intersection sizes displayed in black above each bar. A comparison to previously identified m⁶A sites using the orthogonal technique, miCLIP is included: orange and blue bar fractions show the number of sites within each set intersection that have or do not have an miCLIP peak (*Parker et al., 2020*) within 5 nt, respectively. Percentage of intersections with miCLIP support is displayed in orange above each bar. A small number of sites are significant in the three-way comparison, but in neither two-way comparison (far left bar). (**C–D**) Detection of RNAs immunoprecipitated from Col-0 and *fio1* mutant alleles with anti-m⁶A antibodies using RT-qPCR analysis. The data are presented as strip-plots with mean and 95% CIs, showing the enrichment of U6 and U2 snRNAs over input using (**C**) Synaptic systems #202 003 anti-m⁶A antibody and RNA purified from Col-0 and *fio1-1*, and (**D**) Millipore ABE572 anti-m⁶A antibody and RNA purified from Col-0, *fio1-1*, *fio1-3*, and *fio1-4*. Y axes show $-\Delta C_t$ (m⁶A-IP — input) corrected for input dilution factor. Strip-plots show mean values for three or four independent RT-qPCR amplifications on each biological replicate immunopurification experiment.

The online version of this article includes the following source data and figure supplement(s) for figure 2:

**Source data 1.** LC:MS/MS data for Col-0, *fio1-1*, and *fip37-4* poly(A)+RNA.

**Source data 2.** Differential modification sites detected from a three-way omparison of Col-0, *fip37-4*, and *fio1-1* mutants using nanopore direct RNA sequencing (DRS) data.

**Source data 3.** m⁶A-IP quatitative PCR (qPCR) data for U6 and U2 snRNAs in Col-0, *fio1-1*, *fio1-3, and fio1-4* mutants.

**Source data 4.** Differential poly(A) site usage results for *fip37-4* and *fio1-1* identified using nanopore direct RNA sequencing (DRS) data.

**Figure supplement 1.** Analysis of RNA modification sites detected by nanopore direct sequencing of RNA purified from *fip-37–4* and *fio1-1*.

*Figure 2 continued on next page*

*Figure 2 continued*

**Figure supplement 2.** Identification of modified bases in RNA encoding *Arabidopsis* S-adenosylmethionine (SAM) synthetases using nanopore direct RNA sequencing analysis.

**Figure supplement 3.** (**A–B**) Relative expression of U6 snRNA in (**A**) Col-0 and the *fio1-1* mutant, and (**B**) Col-0, *fio1-1*, *fio1-3*, and *fio1-4* mutants, measured by RT-quantitative (qPCR), compared to U2 snRNA. The means of three or four technical replicates are shown for each biological replicate. Grey bars with points represent mean and 95% CIs.

FIO1 function. In contrast, we identified significant depletion of U6 snRNA in the anti-m⁶A immunopurified RNA from *fio1-1*, *fio1-3*, and *fio1-4* alleles (***Figure 2C–D***). These data are consistent with recent reports (***Wang et al., 2022***; ***Xu et al., 2022***) indicating that *Arabidopsis* U6 snRNA is m⁶A-modified at the conserved position of the AC$^{m6}$**A**GA box and that this modification requires active FIO1.

## Widespread disruption of pre-mRNA splicing in *fio1* mutants

Having found that FIO1 is required for U6 snRNA m⁶A modification, we examined changes in gene expression and pre-mRNA splicing in *fio1* mutants. Nanopore DRS is insightful for mapping the complexity of RNA processing and modification, but throughput currently limits the statistical power to detect splicing changes, and basecalling errors complicate the alignment of exon/intron boundaries (***Parker et al., 2021b***). Consequently, we used Illumina RNA-seq of poly(A)+RNA to analyse gene expression and splicing. Since FIO1 affects the splicing of *MAF2*, which is sensitive to changes in temperature, we included plants grown at 20°C and shifted to either 4, 12, or 28°C for 4 hr prior to harvesting and poly(A)+RNA purification. We sequenced six biological replicates each of wild-type Col-0 and *fio1-3*, generating a minimum of 46 million paired end reads, 150 bp in length, per replicate. On average, 97.9% of read pairs were mappable per replicate, resulting in a total of 3.6 billion mapped read pairs. A summary of the sequencing read statistics is given in ***Supplementary file 1***. The RNA-seq data clearly reveal disruption of *FIO1* gene expression in *fio1-3* (***Figure 3—figure supplement 1A***).

To detect cryptic splice sites that might be activated in *fio1* mutants, we built a condition-specific reference transcriptome derived from the Col-0/*fio1-3* Illumina RNA-seq and Col-0/*fio1-1* nanopore DRS data using the software tool Stringtie2 (***Kovaka et al., 2019***). We quantified the expression of these transcripts using Salmon (***Patro et al., 2017***), then used SUPPA2 to calculate event-level percentage changes in splicing, which also classifies different types of splicing event (***Trincado et al., 2018***). These percent spliced indices (PSIs) were used to fit linear models for each splicing event to identify changes in splicing dependent on temperature, genotype, and temperature×genotype interactions. Analysis of the Col-0 temperature-shift data served as a control. Consistent with previous studies (***Calixto et al., 2018***), we found that intron retention is the predominant class of alternative splicing event detected when *Arabidopsis* is subjected to different temperatures (***Figure 3A***). In contrast, when we repeated this analysis to classify alternative splicing events that differ between *fio1-3* and Col-0, we found that a larger proportion of alternative splicing events were classified as alternative 5′SS usage – 34.4%, compared to only 18.6% of temperature-dependent events (***Figure 3B***, ***Figure 3—figure supplement 1B***). In addition, we detected changes in the PSI of retained introns, exon skipping, and alternative 3′SS selection (***Figure 3B***, ***Figure 3—figure supplement 1B***). There was a significant overlap between the alternative splicing events that were sensitive to loss of *FIO1*, and those that were sensitive to temperature (hypergeometric-test p<1 × 10⁻¹⁶). However, in 64.2% of *fio1*-sensitive events, loss of *FIO1* did not alter splicing responses to temperature (***Figure 3B***). Of the remaining 2505 splicing events that did have altered temperature sensitivity in the absence of FIO1, 38.4% were alternative 5′SSs, and of these, 69.9% had greater sensitivity to loss of *FIO1* at 28°C than at 4°C. This suggests that canonical 5′SS selection at elevated temperature requires FIO1 (***Figure 3—figure supplement 1C***).

Analysis of the RNA-seq data confirmed the increased retention of *MAF2* intron 3 in *fio1-3*, consistent with the results of our two-step mutant screen (***Figure 3C***). Loss of *FIO1* decreased the splicing efficiency of *MAF2* intron 3 by approximately equivalent amounts at all temperatures (***Figure 3D***), implying that FIO1 is not required to generate *MAF2* intron 3 temperature sensitivity.

We detected changes in gene expression consistent with the early flowering phenotype of *fio1* mutant alleles (***Figure 3—figure supplement 2A–G***). For example, *FT* and *SOC1* mRNA levels were increased in *fio1-3* (***Figure 3—figure supplement 2A–B***). In contrast, the mRNA levels of the floral

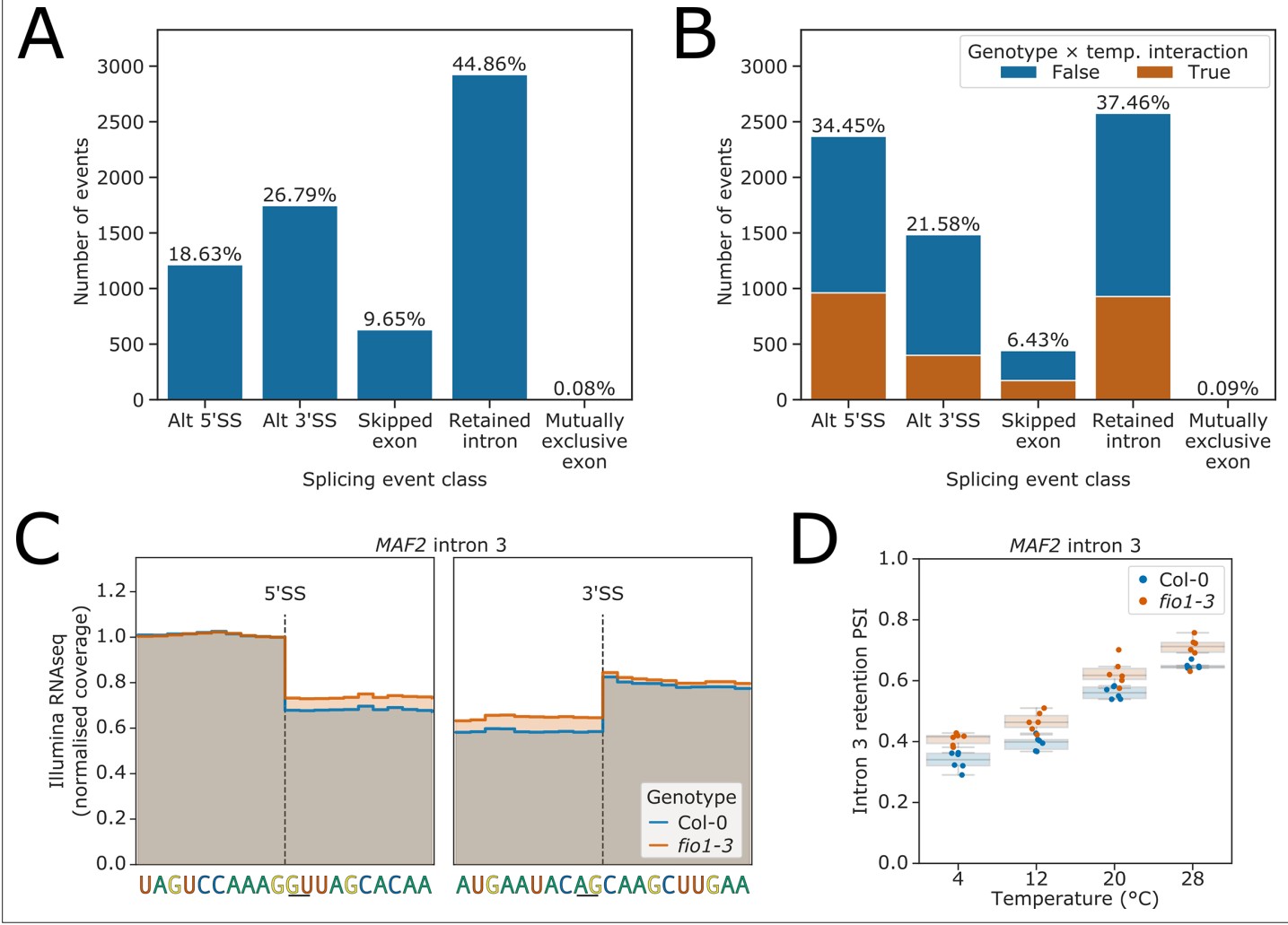

**Figure 3.** Splicing events sensitive to temperature and loss of *FIO1*. (**A–B**) Analysis of Illumina RNA-seq data presented as bar plots showing the proportion of splicing events of each class, as labelled by SUPPA, which have significantly different usage (false discovery rate [FDR] <0.05) at either (**A**) varying temperatures or (**B**) in *fio1-3*. In (**B**), events for which the response to temperature changes in *fio1-3* are shown in orange. (**C**) Gene track of Illumina RNA-seq reads showing the change in retention of *MAF2* intron 3 in *fio1-3*, at 20°C. Expression is normalised by the read coverage at the −1 position of the 5' splice site (5'SS). (**D**) Boxplot of Illumina RNA-seq analysis showing the change in retention of *MAF2* intron 3 at varying temperatures and in *fio1-3*.

The online version of this article includes the following source data and figure supplement(s) for figure 3:

**Source data 1.** Differential gene expression analysis results from Illumina RNA-seq experiment on Col-0 and *fio1-3* mutants at four temperatures.

**Source data 2.** Differential splicing analysis results from Illumina RNA-seq experiment on Col-0 and *fio1-3* mutants at four temperatures.

**Figure supplement 1.** Analysis of Illumina RNA-seq data.

**Figure supplement 2.** Illumina RNA-seq analysis of differential gene expression between Col-0 and *fio1-3*.

repressors *FLM*, *MAF2*, *MAF3*, *MAF4*, *MAF5*, and *FLC* were reduced (*Figure 3—figure supplement 2C–H*). In the case of *FLC*, both sense and antisense transcript levels were reduced in *fio1-3* (*Figure 3—figure supplement 2H–I*), but there were no detectable changes in splicing patterns or in the alternative polyadenylation of the antisense transcripts (*Figure 3—figure supplement 2J*). In these experimental conditions, loss of FIO1 did not affect RNA-level expression or splicing of either *CO* or *SVP* (*Figure 3—figure supplement 2K–L*). In contrast, *fio1-3* mutants exhibited defects in the splicing of pre-mRNA encoding MAF2, MAF3, FLM, and other genes that influence flowering time, as detailed further below.

Overall, we conclude that the predominant molecular phenotype resulting from loss of FIO1 function is a major disruption to patterns of splicing in a manner that is mostly independent of temperature. To understand the basis of these splicing differences, we analysed each of these splicing classes separately.

## Alternative selection of 5'SSs in *fio1-3* can be explained by U6 and U5 snRNA target sequences

We identified 2369 changes in 5'SS choice caused by loss of FIO1 function. Alternative 5'SSs were almost equally likely to be selected upstream or downstream, with many examples of 5'SS shifts of exactly −4 nt,+4 nt, and +5 nt (*Figure 4—figure supplement 1A*). We asked if differences in *cis*-element features could account for FIO1-dependent 5'SS selection. Comparing 5'SSs that exhibited reduced selection in *fio1-3* with their corresponding alternative 5'SSs, which were more frequently selected in *fio1-3*, we found a significant difference in base composition at the −2 to +5 positions (G-test $p<1 \times 10^{-16}$). Specifically, we identified a //GURAG motif (R=A or G) at 5'SSs sensitive to the loss of FIO1 function (*Figure 4A*). This motif appeared more commonly in *fio1*-sensitive 5'SSs than in the full set of 5'SSs observed in the RNA-seq dataset (*Figure 4—figure supplement 1B*) and is consistent with recognition by the U6 snRNA ACAGA box (*Kandels-Lewis and Séraphin, 1993*; *Lesser and Guthrie, 1993*; *Sawa and Abelson, 1992*). In contrast, the alternative 5'SSs selected more frequently in *fio1-3* were characterised by an AG//GU motif (*Figure 4B*). This motif is consistent with U5 snRNA loop 1 recognition of the upstream exon sequences (*Galej et al., 2016*; *Newman and Norman, 1992*). There was no significant difference between the positional base frequencies for 5'SSs that were preferred at lower and elevated temperatures in Col-0 (G-test p=0.28; *Figure 4—figure supplement 1C–D*), demonstrating that altered 5'SS sequence preference is specific to the *fio1-3* mutant and not a general feature of alternative splicing.

The widespread changes in 5'SS selection are evident at individual loci. For example, at *AtSAR1* (AT1G33410), loss of FIO1 function, at 20°C, results in a 52.2% reduction in the use of the normally near constitutive 5'SS in intron 21, UC//GUGAG, with a reciprocal increase in the use of a cryptic 5'SS UG//GUAUU 26 nt downstream (*Figure 4C and D*). At elevated temperatures, the selection of this cryptic 5'SS becomes increasingly preferred (*Figure 4D*). The change in usage of this 5'SS is confirmed in the *fio1-1* allele using orthogonal nanopore DRS (*Figure 4E*). We could also identify cryptic alternative 5'SSs in the second intron of *MAF2* (*Figure 4—figure supplement 2A–B*), and the third exon of another floral repressor, *FLM*, used in *fio1-3* (*Figure 4—figure supplement 2C–D*). The overall effect of these (and other) splicing changes is to reduce the fraction of productive *AtSAR1*, *MAF2*, and *FLM* transcripts in *fio1-3* (*Figure 4—figure supplement 2E–G*). Finally, at the gene encoding the METTL3-like writer complex component MTB (the orthologue of human METTL14), loss of FIO1 function resulted in a 40.4% increase in the use of a cryptic 5'SS in intron 4 that introduces a premature termination codon into the *MTB* open reading frame (*Figure 4—figure supplement 2H–I*). Consequently, loss of *FIO1* may have indirect effects on $m^6A$ deposition by the METTL3-like writer complex.

Remarkably, the presence or absence of an A at the +4 position ($A_{+4}$) correctly separates 78.6% of the 5'SSs exhibiting decreased and increased usage in *fio1-3* (*Figure 4F*). In humans, where pairing of the 5'SS to U6 snRNA occurs before activation of the spliceosome, the 5'SS $A_{+4}$ faces the $m^6A$ of the U6 snRNA $AC^{m6}\mathbf{A}GA$ box in the B complex before docking of the 5'SS in the active site (*Bertram et al., 2017*). Of the 5'SSs with increased usage in *fio1-3*, 48.2% have $U_{+4}$, which could make a Watson–Crick base pair with the corresponding unmethylated residue of U6 snRNA. In total, 61.8% of 5'SS changes in *fio1-3* is associated with a switch from $A_{+4}$ to $B_{+4}$ (B=C, G or U), of which 58.8% are $A{\rightarrow}U_{+4}$ (*Figure 4—figure supplement 3A*). In comparison, only 3.8% of alternative 5'SS pairs are reciprocal $B{\rightarrow}A_{+4}$ switches, indicating that this shift is strongly directional. A further 8.0% of alternative 5'SS pairs are $S{\rightarrow}U_{+4}$ (S=C or G), suggesting that a Watson–Crick A–U base-pair is favoured when U6 snRNA is not $m^6$-modified. Surprisingly, 22.3% of alternative 5'SS pairs have the same base at the +4 position (*Figure 4—figure supplement 3A*). However, of these 5'SS pairs, the majority are associated with a $G{\rightarrow}H_{+5}$ (H=A, C or U) and/or $H{\rightarrow}G_{-1}$ switch that weakens interactions with the U6 snRNA ACAGA box and strengthens U5 snRNA loop 1 interactions (*Figure 4—figure supplement 3B*).

We next examined how the effect size (absolute ΔPSI) of splicing changes correlated with the base at the +4 position of the 5'SS. The largest effect sizes were associated with $A{\rightarrow}U_{+4}$ (*Figure 4G*). In contrast, alternative 5'SS pairs with an $A{\rightarrow}A_{+4}$ shift had smaller effect sizes. We found that

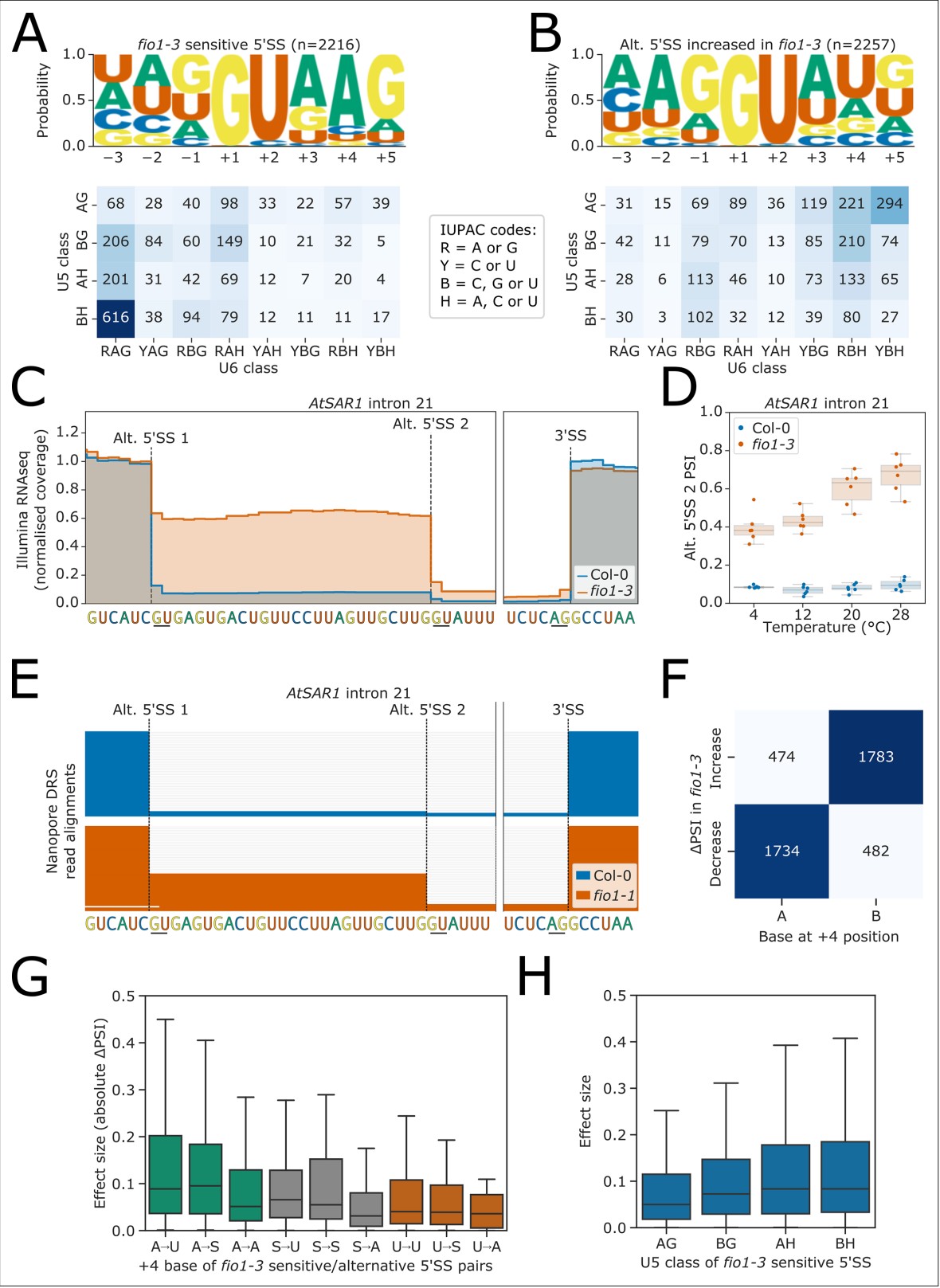

**Figure 4.** Effect of *fio1-3* on alternative 5′ splice sites. (**A–B**) Sequence logos and heatmap showing the distribution of U5 snRNA and U6 snRNA interacting sequence classes for 5′ splice sites (5′SSs) which (**A**) are sensitive to loss of *FIO1* function or (**B**) have increased usage in *fio1-3* based on the analysis of Illumina RNA-seq data. Motifs are shown for the −3 to +5 positions of the 5′SS. U5 classes are based upon the distance of the −2 to −1 positions of the 5′SS from the consensus motif AG. U6 classes are based upon the distance of the +3 to +5 positions of the 5′SS from the consensus

*Figure 4 continued on next page*

*Figure 4 continued*

motif RAG. (**C**) Gene track of Illumina RNA-seq reads showing alternative 5'SS usage at *AtSAR1* intron 21 in *fio1-3*, at 20°C. Expression is normalised by the read coverage at the −1 position of the 5'SS. (**D**) Illumina RNA-seq analysis visualised with a boxplot showing the change in usage of the cryptic alternative 5'SS (Alt 5'SS 2) in *AtSAR1* intron 21 at varying temperatures in Col-0 and *fio1-3*. (**E**) Gene track showing alternative 5'SS usage at *AtSAR1* intron 21 in *fio1-1*, identified using nanopore direct RNA sequencing (DRS) read alignments. Alignments have been subsampled to a maximum of 50 per condition. (**F**) Contingency table showing the relationship between the nucleotide at the +4 position, and the direction of change in 5'SS usage in *fio1-3*, for pairs of alternative 5'SSs with significantly altered usage in *fio1-3* analysed in Illumina RNA-seq data. (**G**) Boxplot showing effect sizes of pairs of alternative 5'SSs with significantly altered usage in *fio1-3*, separated by +4 position bases (A→U indicates that 5'SS with reduced usage has $A_{+4}$, 5'SS with increased usage has $U_{+4}$). (**H**) Boxplot showing effect sizes of pairs of alternative 5'SSs with significantly altered usage in *fio1-3*, separated by U5 classification.

The online version of this article includes the following source data and figure supplement(s) for figure 4:

**Source data 1.** Differential productive transcription/Nonsense mediated RNA decay (NMD) analysis results from Illumina RNA-seq experiment on Col-0 and *fio1-3* mutants at four temperatures.

**Figure supplement 1.** (**A**) Histogram showing the distance between alternative 5' splice site (5'SS) pairs with significantly different usage in *fio1-3* detected by Illumina RNA-seq analysis.

**Figure supplement 2.** (**A**) Gene track of Illumina RNA-seq reads showing alternative 5' splice site (5'SS) usage at *MAF2* intron 2 in *fio1-3*, at 20°C.

**Figure supplement 3.** (**A**) Contingency table showing the relationship between the bases at the 4+ position for *fio1-3*-sensitive 5' splice sites (5'SSs) with reduced usage in *fio1-3* and alternative 5'SSs with increased usage in *fio1-3* revealed through Illumina RNA-seq data analysis.

---

*fio1-3*-sensitive 5'SSs with 5'SS $G_{+5}$ had larger effect sizes but that $G_{+5}$ at the alternative 5'SS had less effect, suggesting that $G_{+5}$ is only deleterious in *fio1-3* when in combination with $A_{+4}$ (***Figure 4—figure supplement 3C***). Finally, we found that *fio1-3*-sensitive 5'SSs with AG//GU motifs had smaller effect sizes, indicating that favourable interactions with U5 snRNA loop 1 can suppress the effect of unfavourable U6 snRNA interactions in *fio1-3* (***Figure 4H***).

In summary, these analyses demonstrate widespread change in 5'SS selection in *fio1* mutants. The 5'SSs with a strong match to the U6 snRNA ACAGA box and/or an $A_{+4}$ are most sensitive to loss of FIO1. This sensitivity can be suppressed by a strong match in the 5' exon to U5 snRNA loop 1. Alternative 5'SSs that are used more in *fio1* have $U_{+4}$, as well as 5' exon sequence features that favour recognition by U5 snRNA loop 1.

## FIO1-dependent changes in intron retention and exon skipping can be explained by U6 and U5 snRNA target sequences

In addition to altered 5'SS selection, we detected many instances of intron retention and exon skipping in *fio1-3*. We therefore asked if these FIO1-sensitive splicing phenotypes were associated with specific sequence motifs.

We identified 2576 introns with altered levels of retention in *fio1-3*, of which 55.8% had increased retention, and 44.2% had decreased retention (i.e. more splicing). Analysis of 5'SS sequences at introns with increased retention indicates that 76.9% had $A_{+4}$, as well as a weaker interaction potential with U5 snRNA loop 1 (***Figure 5A***). For example, at *WNK1* (AT3G04910), we detected increased retention of intron 5 (CG//GUGAG) in *fio1-3* (***Figure 5B–C***) and *fio1-1* (***Figure 5D***) using Illumina RNA-seq and nanopore DRS analysis, respectively. This indicates that these introns are normally recognised by strong U6 snRNA interactions and are less efficiently spliced in the absence of *FIO1*-dependent U6 snRNA $m^6A$ modification. The remaining introns with significantly altered retention are more efficiently spliced in *fio1-3*. These introns tend to have stronger matches to U5 snRNA loop 1 in the 5' exon, and only 34.7% have $A_{+4}$, whilst 44.3% have $U_{+4}$ (***Figure 5A***). This suggests that loss of $m^6A$ from U6 snRNA actually increases the splicing efficiency of introns with U at the +4 position of the 5'SS.

A common form of alternative splicing involves the excision of an exon and flanking introns in a splicing event called exon skipping. These events involve two 5'SSs: those at the upstream intron and those at the downstream intron (i.e. at the 3' end of the skipped exon; ***Figure 5E–F***). We identified 442 exons with significantly different levels of skipping in *fio1-3*. Of these, 53.6% have increased levels of inclusion and 46.4% have increased levels of skipping.

In cases where exon skipping increased in *fio1-3*, more of the 5'SSs at downstream introns had $A_{+4}$ compared to the 5'SSs at upstream introns. Conversely, more of the 5'SSs at upstream introns had strong U5 snRNA loop 1 interacting sequences (***Figure 5E***, ***Figure 5—figure supplement 1A***). This is consistent with a relative weakening of the recognition of the 5'SS at the downstream intron in *fio1-3*.

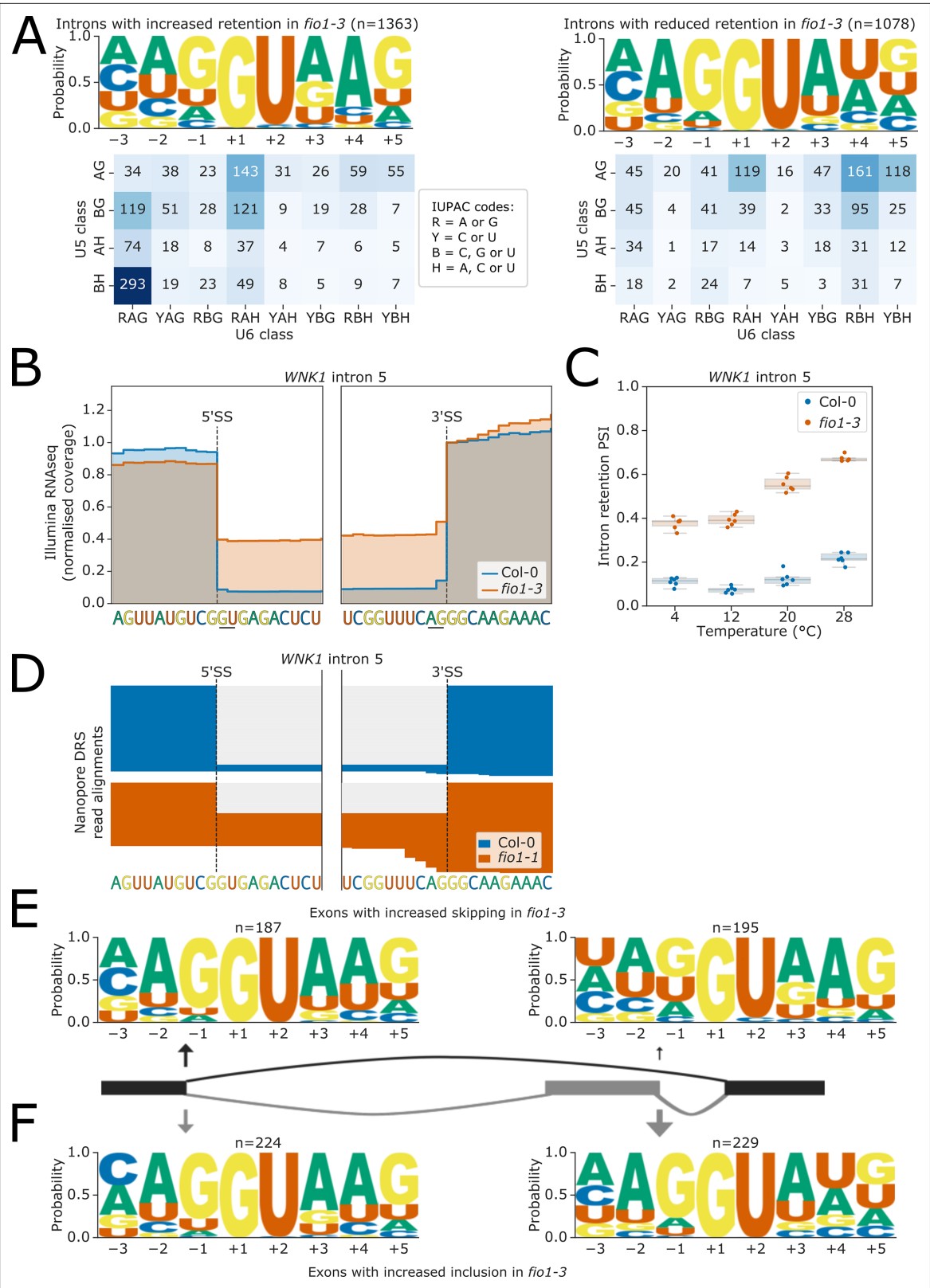

**Figure 5.** Effect of *fio1-3* on retained introns and exon skipping events. (**A**) Sequence logos and heatmaps showing the distribution of U5 snRNA and U6 snRNA interacting sequence classes for 5′ splice sites (5′SSs) which have increased (left) and decreased (right) retention in *fio1-3* based on Illumina RNA-seq data analysis. Motifs are shown for the −3 to +5 positions of the 5′SS. U5 classes are based upon the distance of the −2 to −1 positions of the 5′SS from the consensus motif AG. U6 classes are based upon the distance of the +3 to +5 positions of the 5′SS from the consensus motif RAG. (**B**) Gene

*Figure 5 continued on next page*

*Figure 5 continued*

track of Illumina RNA-seq reads showing intron retention at *WNK1* intron 5 in *fio1-3,* at 20°C. Expression is normalised by the read coverage at the +1 position of the 3' splice site (3'SS). (**C**) Boxplot of Illumina RNA-seq data analysis showing the change in intron retention of *WNK1* intron 5 at varying temperatures, in Col-0 and *fio1-3*. (**D**) Gene track showing intron retention at *WNK1* intron 5 in *fio1-1,* identified using nanopore direct RNA sequencing (DRS) read alignments. Alignments have been subsampled to a maximum of 50 per condition. (**E–F**) Sequence logos for 5'SSs at introns upstream (left) and downstream (right) of exons with (**E**) increased skipping or (**F**) increased retention in *fio1-3* based on Illumina RNA-seq data analysis.

The online version of this article includes the following figure supplement(s) for figure 5:

**Figure supplement 1.** (**A**) Heatmaps showing the distribution of U5 snRNA and U6 snRNA interacting sequence classes for 5' splice site (5'SSs) of exons (right) and upstream 5'SSs (left) which have increased (above) and decreased (below) skipping in *fio1-3* as deduced from analysis of Illumina RNA-seq data.

For example, at the gene encoding the floral repressor MAF3 (AT5G65060), there is an increase in skipping of exon 2 in *fio1-3* (***Figure 5—figure supplement 1B–C***). The 5'SS at the 3' end of this exon is UA//GUAAG, whereas the upstream 5'SS (AA//GUAAG) is a stronger match to U5 snRNA loop 1.

In cases where exon inclusion increased in *fio1-3*, we found that the majority of 5'SSs at upstream introns had $A_{+4}$, whereas a plurality of 5'SSs at downstream introns had $U_{+4}$ (***Figure 5F***, ***Figure 5— figure supplement 1A***). For example, at *PTB1* (AT3G01150), there is an increase in inclusion of exon 3 in *fio1-3* (***Figure 5—figure supplement 1D–E***). The 5'SS at the 3' end of this exon is AG//GUGUC, whereas the upstream 5'SS is UG//GUGAG. Although both the upstream and downstream 5'SSs are used when the exon is retained, it is possible that the relative strengthening of recognition of the downstream 5'SS in *fio1-3* improves exon definition.

In summary, we can identify and characterise changes in intron retention and exon skipping sensitive to loss of FIO1 function. Although the outcome of these splicing events is different from alternative 5'SS selection, they are associated with the same changes in 5'SS sequence as those associated with alternative 5'SS selection.

## Alternative 3'SS usage in *fio1-3* can be explained by U6 snRNA-dependent interactions between 5' and 3'SSs

We observed 1484 instances where 3'SS selection was altered in *fio1-3*. In some cases, these alternative 3'SSs were linked with altered 5'SS choice or intron retention, but this accounted for a relatively small proportion of events (***Figure 6—figure supplement 1A***). Alternative 3'SS selection was equally likely to switch in an upstream or downstream direction in *fio1-3* (***Figure 6A***). There was a strong enrichment for very local switching of 3'SSs, with 37.9% of alternative 3'SSs occurring within 6 nt of the *fio1-3*-sensitive 3'SS, and 18.9% of examples occurring at exactly 3 nt upstream or downstream. These examples correspond to NAGNAG-like acceptors, which have previously been characterised in multiple species including human and *Arabidopsis* (***Bradley et al., 2012***; ***Hiller et al., 2004***; ***Schindler et al., 2008***).

To examine the 5'SSs associated with alternative 3'SSs used in *fio1-3*, we separated examples with increased upstream and downstream 3'SS usage and performed motif analysis. At introns with a relative increase in upstream 3'SS usage in *fio1-3*, we found that 80.5% of the corresponding 5'SSs were characterised by $A_{+4}$ (***Figure 6B***). For example, at *LHY* (AT1G01060), we identified an AA//GUAAG 5'SS and a UAG\\CAG\\ alternative 3'SS pair in intron 7 (***Figure 6D***). In *fio1-3*, at 20°C, there is a shift in favour of the upstream UAG\\ 3'SS (***Figure 6E***). This 3'SS switch is supported by orthogonal nanopore DRS analysis of the *fio1-1* allele (***Figure 6F***). Conversely, when we analysed the features of introns where a relative increase in downstream 3'SS usage was detected in *fio1-3*, we found that only 37.0% of these 5'SSs had $A_{+4}$, whereas 48.8% had $U_{+4}$ (***Figure 6C***). For example, at *MAF2* intron 4, we identified an AG//GUAUU 5'SS and a UAG\\ACAG\\ alternative 3'SS pair (***Figure 6—figure supplement 1B***). In *fio1-3*, at 20°C, there is a shift in favour of the downstream CAG\\ 3'SS (***Figure 6—figure supplement 1C***).

3'SS choice involves scanning downstream from the branchpoint to the first available 3'SS motif (***Smith et al., 1993***). However, competition with downstream 3'SSs can occur within a short range, such as in NAGNAG acceptors. Downstream 3'SSs in *fio1*-sensitive alternative 3'SS pairs were more likely than upstream 3'SSs to have a cytosine as the −3 position (***Figure 6B–C***). However, this appears to reflect differences in the background rate of $C_{-3}$ upstream and downstream (***Figure 6—figure supplement 1D***), rather than a change in 3'SSs motif competitiveness in *fio1-3* (***Bradley et al., 2012***;

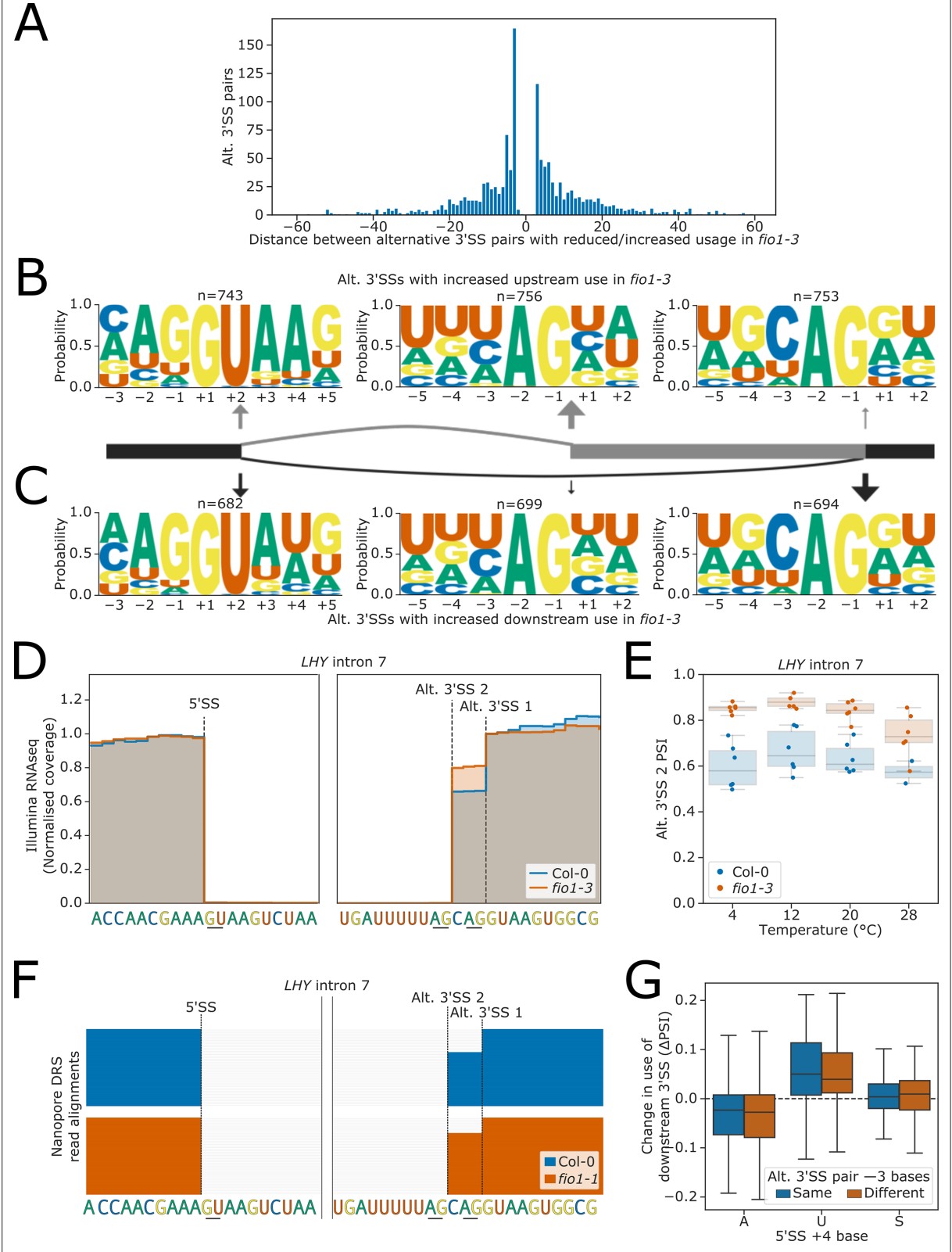

**Figure 6.** Effect of *fio1-3* on alternative 3′ splice site (3′SS) usage. (**A**) Histogram showing the distance between alternative 3′SS pairs with significantly different usage in *fio1-3* revealed by Illumina RNA-seq data analysis. Negative distances represent shifts toward greater usage of upstream 3′SSs, whilst positive distances represent shifts toward greater usage of downstream 3′SSs. (**B–C**) Sequence logos for 5′ splice sites (5′SSs; left), upstream 3′SSs (middle), and downstream 3′SSs (right) at pairs of alternative 3′SSs with increased (**B**) upstream or (**C**) downstream usage in *fio1-3* revealed by

*Figure 6 continued on next page*

*Figure 6 continued*

Illumina RNA-seq data analysis. 5'SS logos are for −3 to +5 positions, and 3'SS logos are for −5 to +2 positions. (**D**) Gene track of Illumina RNA-seq reads showing alternative 3'SS usage at *LHY* intron 5 in *fio1-3*, at 20°C. Expression is normalised by the read coverage at the −1 position of the 5'SS. (**E**) Boxplot of Illumina RNA-seq data analysis showing the change in usage of the upstream alternative 5'SS (Alt 5'SS 2) in *LHY* intron 5 at varying temperatures in Col-0 and in *fio1-3*. (**F**) Gene track showing intron retention at *LHY* intron 5 in *fio1-1*, identified using nanopore direct RNA sequencing (DRS) read alignments. Alignments have been subsampled to a maximum of 50 per condition. (**G**) Boxplot of Illumina RNA-seq data analysis showing the change in usage of downstream 3'SSs in alternative 3'SS pairs with different 5'SS +4 bases, separated by whether the base at the −3 position of the two alternative 3'SSs is the same (e.g. CAG\\CAG\\) or different (e.g. UAG\\CAG\\).

The online version of this article includes the following figure supplement(s) for figure 6:

**Figure supplement 1.** (**A**) Upset plot derived from Illumina RNA-seq data analysis showing the overlap of *fio1*-sensitive 5' splice sites (5'SSs) involved in alternative 5'SS usage, with 5'SSs at *fio1*-sensitive intron retention events and alternative 3' splice sites (3'SSs).

**Figure supplement 2.** (**A**) Line-plot showing GC content of Illumina RNA-seq reads from *Wang et al., 2022*. One replicate of Col-0 data was discarded due to extreme negative GC-bias likely resulting from PCR overamplification. (**B**) Sequence logos for 5' splice sites (5'SSs) identified from RNA-seq data taken from *Wang et al., 2022*, which (above) are sensitive to loss of FIO1 function or (below) have increased usage in *fio1-1*.

**Figure supplement 3.** (**A–B**) Sequence logos for 5' splice sites (5'SSs) identified from Illumina RNA-seq data of (**A**) *fio1-1* and (**B**) *fio1-5* alleles reanalysed from *Sun et al., 2022*, which (left) are sensitive to loss of FIO1 function or (right) have increased usage in the respective *fio1* mutants. (**C**) Upset plot showing the intersection of the sets of alternative 5'SS events which are identified from Illumina RNA-seq data of *fio1-1* and *fio1-5* alleles from *Sun et al., 2022* independent RNA-seq of the *fio1-3* allele from this study.

**Figure supplement 4.** (**A–F**) Gene tracks showing (**A and D**) alternative 5' splice site (5'SS) selection at *AtSAR1* intron 21, (**B and E**) intron retention at *WNK1* intron 5, and (**C and F**) alternative 5'SS selection at *MTB* intron 4 in (**A–C**) the *fio1-2* mutant, identified using nanopore direct RNA sequencing (DRS) read alignments reanalysed from *Xu et al., 2022* and (**D–F**) *fio-1* and *fio1-5* mutants, identified using nanopore DRS read alignments reanalysed from *Sun et al., 2022*.

*Smith et al., 1993*). To analyse whether the −3 position contributes to changes in alternative 3'SS usage in *fio1*, we separated *fio1*-sensitive 3'SS pairs where the base at the −3 position was the same at both the upstream and downstream 3'SS, from examples which had different bases at the −3 position. We found that among both sets of 3'SS pairs, 5'SS $A_{+4}$ was still associated with increased upstream usage in *fio1-3*, whereas $U_{+4}$ was associated with increased downstream usage (*Figure 6G*). This demonstrates that changes in 3'SS usage in *fio1-3* are caused by a change in the competitiveness of distal 3'SSs, irrespective of 3'SS motif.

These findings link selection of the 3'SS and 5'SS. When U6 snRNA recognition of the 5'SS is favoured by either an $m^6A:A_{+4}$ interaction in WT Col-0 or $A:U_{+4}$ interaction in *fio1-3*, this increases the usage of distal 3'SSs. When the U6 snRNA/5'SS interaction is less favoured, then upstream 3'SSs are more likely to be used. We conclude that the interactions of U6 snRNA with the 5'SS can influence usage of competing 3'SSs.

## Reanalysis of RNA-seq data from independent *fio1* alleles confirms disruption to splicing

In contrast to our findings, three recently published studies of *fio1* mutants failed to identify an impact on splicing (*Wang et al., 2022*; *Xu et al., 2022*; *Sun et al., 2022*). We reanalysed these published data to understand the basis for this difference. Two studies performed Illumina RNA-seq analysis (*Wang et al., 2022*; *Sun et al., 2022*) but were not well powered with only two replicates per condition. The Illumina RNA-seq data from Wang et al. were of poor quality, with low mapping rates and one unusable replicate due to PCR-bias (*Figure 6—figure supplement 2A*). Nevertheless, we were able to detect 384 alternative 5'SS events by using our condition-specific transcriptome assembly and performing differential splicing analysis with SUPPA2. These events were associated with a shift from 5'SSs with $A_{+4}$ to alternatives with $U_{+4}$ (*Figure 6—figure supplement 2B*). We reanalysed the Illumina RNA-Seq data from Sun et al. (two replicates per condition and 20–26 million reads per replicate) to identify 923 alternative 5'SS events in *fio1-1* and 947 alternative 5'SS events in a CRISPR-induced mutation (*fio1-5*; *Figure 6—figure supplement 3A–B*). There were 675 alternative 5'SS events common to both alleles, and 556 (82.4%) of these were identified in our own analysis of *fio1-3* (*Figure 6—figure supplement 3C*). A third study analysed gene expression changes using nanopore DRS (*Xu et al., 2022*). Using these data, we could confirm splicing changes observed in our own data, for example at *AtSAR1* intron 21, *WNK1* intron 5, and *MTB* intron 4 (*Figure 6—figure supplement 4A–C*). These changes were also visible in nanopore DRS data generated by Sun et al. (*Figure 6—figure supplement 4D–F*).

We conclude that a combination of under-powered datasets and methodological choices in data analysis explain previous failures to link the loss of U6 snRNA methylation in *fio1* mutants to detectable changes in splicing.

### Sequences targeted by U5 and U6 snRNAs are anticorrelated in the 5'SSs of developmentally complex eukaryotes

Our results indicate that a strong match to U5 snRNA loop 1 in the 3' end of the upstream exon offsets the effects of 5'SS $A_{+4}$ unfavourability in *fio1-3*. We, therefore, reasoned that if U5 and U6 cooperate in 5'SS selection, then strong U5 snRNA loop 1 interactions may globally compensate weaker U6 snRNA ACAGA box interactions and vice versa. To test this hypothesis, we used all annotated *Arabidopsis* 5'SS sequences to generate position-specific scoring matrices (PSSMs) for the −2 to −1 positions corresponding to the U5 snRNA loop 1 interacting region, and the +3 to +5 positions corresponding to the U6 snRNA ACAGA box interacting region. We then used these PSSMs to score the U5 and U6 snRNA interaction log-likelihood of each individual 5'SS. 5'SSs with AG//GU motifs have high-scoring U5 PSSM scores, and those with //GURAG motifs have high-scoring U6 PSSM scores. This analysis revealed that U5 and U6 snRNA PSSM scores are indeed negatively correlated in *Arabidopsis* (Spearman's $\rho$ = −0.36, p<1 × 10$^{-16}$). We found that 5'SSs lacking a //GURAG motif had significantly higher U5 PSSM scores than //GURAG 5'SSs (*Figure 7A and B*), whereas BH//GU 5'SSs had significantly higher U6 PSSM scores than AG//GU 5'SSs (*Figure 7C and D*). We found similar anticorrelations in U5 and U6 PSSM scores in other species, including *C. elegans* ($\rho$ = −0.38, p<1 × 10$^{-16}$; *Figure 7—figure supplement 1A*), *Drosophila melanogaster* ($\rho$ = −0.30, p<1 × 10$^{-16}$; *Figure 7—figure supplement 1B*), *Danio rerio* ($\rho$ = −0.36, p<1 × 10$^{-16}$; *Figure 7—figure supplement 1C*), and *Homo sapiens* ($\rho$ = −0.17, p<1 × 10$^{-16}$; *Figure 7—figure supplement 1D*). These findings indicate that there are two major classes of 5'SS in distinct eukaryotes: //GURAG and AG//GU. These two classes occur with approximately equal frequency in the metazoan genomes that we analysed (*Figure 7—figure supplement 1A–D*).

### Opposing U5 and U6 snRNA interaction potential is a feature of alternative 5'SS pairs in some eukaryote genomes

Given that many 5'SSs appear to have either strong U5 or U6 snRNA interacting sequences, we speculated that opposing U5 and U6 snRNA interaction strengths at pairs of alternative 5'SSs could contribute to alternative splicing. To test this hypothesis, we calculated the log odds ratio of U5 and U6 PSSM scores for all pairs of annotated alternative 5'SSs in *Arabidopsis*. We found that these ratios were negatively correlated (Spearman's $\rho$ = −0.25, p<1 × 10$^{-16}$), indicating that alternative 5'SS pairs tend to have complementary strengths with respect to U5 and U6 snRNA recognition (*Figure 7E*). The strength of this complementary relationship varies in different organisms: in humans, we found a stronger negative correlation between relative U5 and U6 PSSM scores ($\rho$ = −0.40, p<1 × 10$^{-16}$; *Figure 7F*), whereas in *C. elegans*, the correlation was weaker ($\rho$ = −0.15, p=8.61 × 10$^{-7}$; *Figure 7—figure supplement 2A*), and in *D. rerio*, there was no correlation ($\rho$ = −0.01, p=0.58; *Figure 7—figure supplement 2B*). We conclude that changing the relative favourability of 5'SSs with stronger U5 or U6 snRNA interacting sequences, as occurs in *fio1-3*, could be a mechanism contributing to alternative 5'SS choice.

## Discussion
### The splicing of mRNAs encoding regulators of flowering time is disrupted in *fio1*

We identified *fio1* through a two-step mutant screen designed to reveal factors that control the splicing of *MAF2* intron 3. Importantly, temperature-sensitive splicing of *MAF2* intron 3 links ambient temperature to flowering time control. *fio1* mutants have increased retention of *MAF2* intron 3 compared to WT Col-0. However, splicing of *MAF2* intron 3 remains responsive to temperature in *fio1* mutants, suggesting that FIO1 is not the thermosensor in this process.

We found that disruption of *FIO1* alters the splicing of mRNA encoding not only *MAF2* but other floral repressors too, including FLM and MAF3. Detectable levels of sense and antisense RNAs at the locus encoding the floral repressor *FLC* were reduced in *fio1*, but no splicing changes were detected.

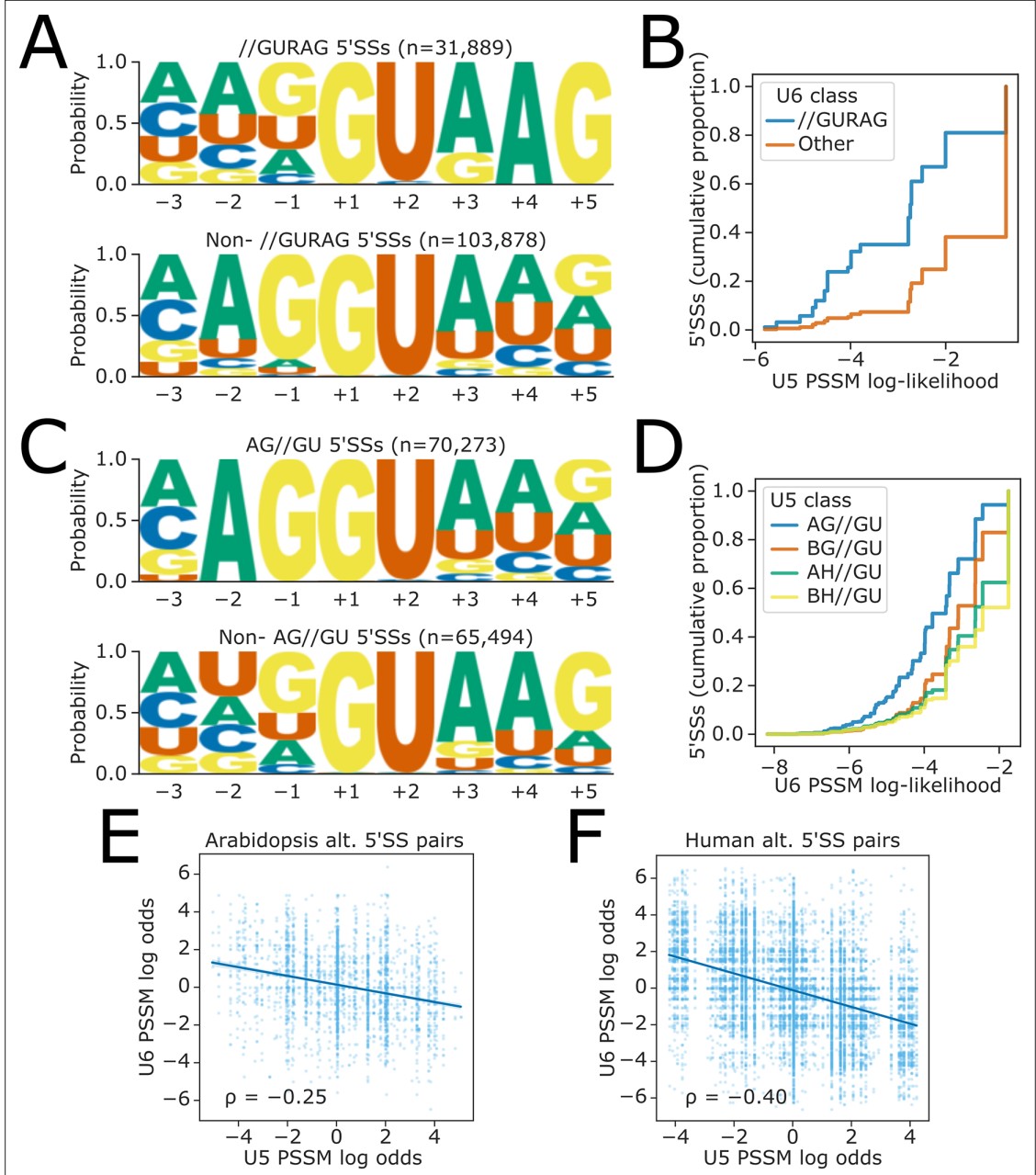

**Figure 7.** Global analysis of U5 and U6 interaction strengths reveals anticorrelation. (**A**) Sequence logos of annotated *Arabidopsis* splice sites showing base frequency probabilities at −3 to +5 positions for (above) 5' splice sites (5'SSs) with //GURAG sequence and (below) all other 5'SSs. (**B**) Empirical cumulative distribution function of *Arabidopsis* U5 position-specific scoring matrix (PSSM) log-likelihood scores for 5'SSs with either //GURAG sequence or all other 5'SSs. U5 PSSM scores are calculated using a PSSM derived from all 5'SSs in the Araport11 reference annotation, at the −2 to −1 positions of the 5'SS, inclusive. (**C**) *Arabidopsis* sequence logos showing base frequency probabilities at the −3 to +5 positions for (above) 5'SSs with AG//GU sequence and (below) all other 5'SSs. (**D**) Empirical cumulative distribution function of *Arabidopsis* U6 PSSM log-likelihood scores for 5'SSs with different U5 classes. U6 PSSM scores are calculated using a PSSM derived from all 5'SSs in the Araport11 reference annotation, at the +3 to +5 positions of the 5'SS, inclusive. (**E–F**) Scatterplot showing the ratio of PSSM log-likelihoods (log-odds ratio) for U5 and U6 snRNA interacting sequences, at pairs of upstream and downstream alternative 5'SSs in (**E**) the *Arabidopsis* Araport11 reference annotation or (**F**) the *H. sapiens* GRCh38 reference annotation. A positive log-odds ratio indicates that the PSSM score of the upstream 5'SS is greater than that of the downstream 5'SS.

The online version of this article includes the following figure supplement(s) for figure 7:

**Figure supplement 1.** (**A, C, E, and G**) Sequence logos showing base frequency probabilities at −3 to +5 positions for (above) 5' splice sites (5'SSs) with //GURAG sequence and (below) all other 5'SSs, in the organisms (**A**) *C.elegans*, (**C**) *D. melanogaster*, (**E**) *D. rerio*, and (**G**) *H. sapiens*.

*Figure 7 continued on next page*

Figure 7 continued

**Figure supplement 2.** (A–B) Scatterplot showing the ratio of position-specific scoring matrix (PSSM) log-likelihoods (log odds ratio) for U5 snRNA and U6 snRNA interacting sequences, at pairs of upstream and downstream alternative 5′ splice sites (5′SSs) in the (A) *C.elegans* or (B) *D. rerio* reference annotation.

FLM, MAF2, MAF3, and FLC function together with SVP in higher-order protein complexes to repress the expression of *FT* and *SOC1* (**Gu et al., 2013**). We detected elevated transcript levels of *FT* and *SOC1* in *fio1* mutants, consistent with the idea that floral repressor activity had been compromised. Splicing changes were present in transcripts encoding circadian modulators such as *LHY* and *WNK1* (**Kumar et al., 2011**; **Mizoguchi et al., 2002**), which may contribute to the lengthening of the circadian period observed in *fio1* mutants (**Kim et al., 2008**). Finally, we found that aberrant splicing limits the functional expression of *AtSAR1* in *fio1* mutants. *AtSAR1* encodes a nucleoporin that controls CO protein abundance (**Dong et al., 2006**; **Li et al., 2020**; **Parry et al., 2006**) and *FLC* expression (**Jung et al., 2013**). As a result, *sar1* mutants are early flowering due to increased CO protein levels (**Li et al., 2020**) and reduced *FLC* expression (**Jung et al., 2013**; **Li et al., 2020**). Overall, these findings suggest that the early flowering phenotype of *fio1* results from splicing changes that reduce the activity of floral repressors and increase the activity of factors that promote flowering, such as CO.

## FIO1-dependent m⁶A modification of U6 snRNA determines splicing accuracy and efficiency

Our data indicate that the major effect of FIO1 occurs through m⁶A modification of U6 snRNA and the subsequent interaction of m⁶A-modified U6 snRNA with target 5′SSs. Recent suggestions that FIO1 does not impact splicing (**Wang et al., 2022**; **Xu et al., 2022**) can be explained by underpowered experiments and unsuitable methodological approaches. Our reanalysis of recently published RNA-seq and nanopore DRS data in independent *fio1* mutant alleles confirms that splicing is indeed disrupted, consistent with the conserved role of FIO1 in methylating U6 snRNA (**Wang et al., 2022**; **Xu et al., 2022**; **Sun et al., 2022**).

We cannot rule out the possibility that FIO1 directly targets other RNA species not detected by our approaches. However, our nanopore DRS analyses did not reveal widespread FIO1-dependent m⁶A sites in poly(A)+mRNA. We found that 8.7% of FIP37-dependent m⁶A sites had slightly altered modification rates in *fio1-1*, suggesting that loss of FIO1 might indirectly affect m⁶A sites written by the METTL3-like complex. Consistent with this, splicing of *MTB* (*METTL14*) RNA is also defective in *fio1* mutants, potentially compromising MTB function. Notably, although data from three recent studies (**Wang et al., 2022**; **Xu et al., 2022**; **Sun et al., 2022**) show almost no overlap in respect of RNAs putatively modified by FIO1-dependent addition of m⁶A (**Sun et al., 2022**), we find that all these datasets confirm the impact of FIO1 on splice site selection.

We can explain the vast majority of *fio1*-sensitive splicing events using simple rules governed by sequence features of 5′SSs. When U6 snRNA is m⁶Amodified, 5′SSs with $A_{+4}$ are favoured. In the absence of U6 snRNA m⁶A, 5′SSs with $U_{+4}$ and/or stronger interactions with U5 snRNA loop 1 are favoured. Cryo-electron microscopy (cryo-EM) structures of the human spliceosome suggest U6 snRNA m⁶A faces 5′SS$_{+4}$ (**Bertram et al., 2017**). Remarkably, we can separate almost 80% of FIO1-dependent 5′SS choices by the identity of the base at the +4 position of 5′SSs alone, suggesting that many of the splicing changes we detect are direct. In contrast, no obvious difference in splice site sequences was associated with temperature-dependent alternative splicing detected in our study.

Our findings are similar to those recently reported for *S. pombe*, where disruption of the FIO1 orthologue MTL16 results in widespread intron retention (**Ishigami et al., 2021**). A more diverse set of splicing events is disrupted in *Arabidopsis fio1* mutants. Nevertheless, in both species, most changes in splicing can be explained by the impact of m⁶A modification on U6 snRNA recognition of the 5′SS$_{+4}$ position or the relative strength of U5 and U6 snRNA interactions at 5′SSs. Defective splicing of specific introns in *S. pombe mtl16Δ* strains can be experimentally rescued by expression of mutated U5 snRNAs designed to strengthen U5 loop 1 interactions with the upstream exon (**Ishigami et al., 2021**). Together, these findings reveal the impact of U6 snRNA m⁶A modification and indicate that U5 and U6 snRNAs have cooperative and compensatory roles in 5′SS selection dependent on 5′SS sequence (**Figure 8A**).

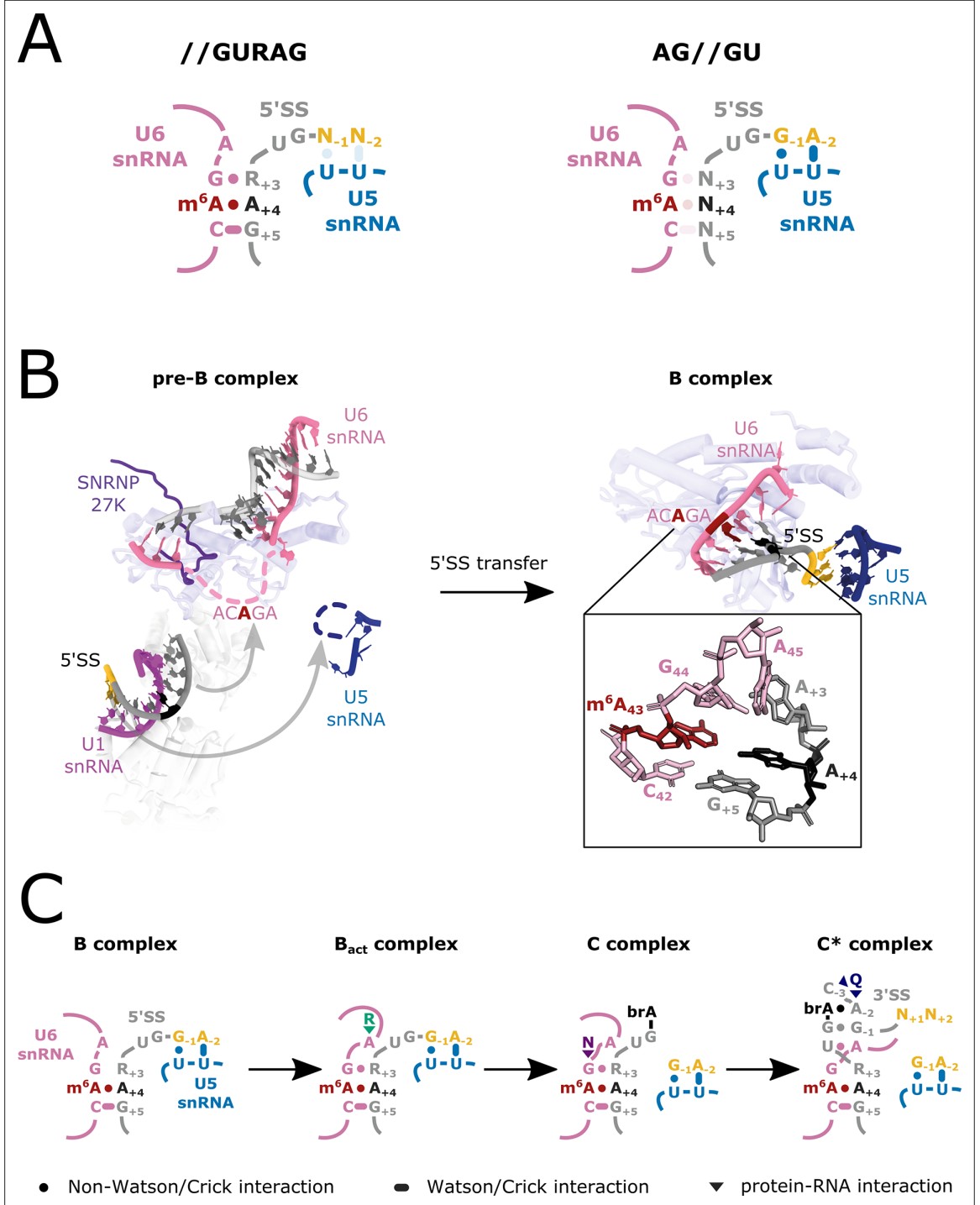

**Figure 8.** U6 m⁶A:5′SSA $_{A+4}$ interactions during splicing. (**A**) Model depicting U5 and U6 snRNA interactions with two major classes of 5′SS: //GURAG and AG//GU 5′SSs. //GURAG 5′SSs form strong interactions with U6 snRNA ACAGA (darkly shaded) and weaker interactions with U5 snRNA loop 1 (lightly shaded). AG//GU 5′SSs form strong interactions with U5 snRNA loop 1 and weaker interactions with U6 snRNA ACAGA. (**B**) Cryo-electron microscopy (cryo-EM) analysis of human pre-B and B complexes with RNA interactions detailed in the expanded section (PDB 6AHD; *Bertram et al., 2017*) and Prp8 shown in the background as a common scaling reference. The U6 snRNA ACAGA and U5 snRNA loop 1 sequences are missing from cryo-EM structures at this stage, probably because they present as flexible loops. In B complex, C42 and 5′SS G₊₅ form a canonical Watson–Crick pair. m⁶A43 and 5′SS A₊₄ form a trans Hoogsteen sugar edge interaction (*Leontis et al., 2002*) that caps and stabilises the U6/5′SS helix by stacking because U6 snRNA G44 and 5′SS A₊₃ have not yet formed a stable interaction. The 5′SS is kinked, and U5 snRNA loop 1 is docked on the upstream exon. The methyl group of U6 snRNA m⁶A43 is not modelled in the structure due to lack of resolution. (**C**) Model depicting U6 m⁶A interactions at different stages of splicing. In B complex, U6 m⁶A43 stabilises the U6/5′SS helix by stacking. As the active site forms in B$_{act}$, this role becomes less important because U6

*Figure 8 continued on next page*

*Figure 8 continued*

snRNA G44 interacts more stably with 5'SS$_{+3}$ and U6 A45 stacks on the helix stabilised by R554 of SF3B2. In C complex, the U6/5'SS helix is stabilised by N57 of hYJU2. In C* complex, the U6m$^6$A43:5'SS$_{+4}$ interaction becomes more important again because the 5'SS$_{+3}$ pivots to a new position. The m$^6$A43 and 5'SS$_{+4}$ pair forms part of a continuous helical stack with the docked 3' splice site (3'SS), which is capped by the interaction between 3'SS$_{-3}$ and Q1522 of Prp8. For more detail, see *Figure 8—figure supplement 1*.

The online version of this article includes the following figure supplement(s) for figure 8:

**Figure supplement 1.** Tracing m$^6$A modified U6 snRNA in cryo-electron microscopy (cryo-EM) structures of different spliceosomal complexes with cryo-EM reveals in pre-B complex (PDB 6QX9, *Charenton et al., 2019*), the ACAGA box is flexible and disordered.

## Two major classes of 5'SS linking cooperative and compensatory roles for U5 and U6 snRNA in 5'SS selection

Our analyses of annotated *Arabidopsis* 5'SSs identified anti-correlated biases in sequence composition at U5 and U6 snRNA interacting positions. We found the same to be true for 5'SSs in other eukaryotes, including humans. Compensatory patterns of base composition at human 5'SSs have been described (*Burge and Karlin, 1997*; *Carmel et al., 2004*; *Sibley et al., 2016*; *Wong et al., 2018*; *Artemyeva-Isman and Porter, 2021*). For example, a 'seesaw linkage' pattern was observed where −1 G permits any nucleotide at position +5, and conversely, +5 G permits any nucleotide at position −1. Our findings suggest that such compensatory base composition at 5'SSs can be accounted for by two major classes of 5'SS recognised mainly by interactions with either U5 snRNA loop 1 or the U6 snRNA AC**A**GA box. Furthermore, the cooperativity of U5 and U6 snRNA function in 5'SS selection may explain why having Gs at both −1 and +5 positions is highly preferential for efficient splicing (*Wong et al., 2018*). Compensatory interactions are likely to facilitate degeneracy in sequences that can be recognised as 5'SSs. Degeneracy may act as a buffer against deleterious mutations in splice site sequences, as well as lowering barriers to the evolution of new splicing structures. This is important because alternative splicing is clearly associated with developmental complexity (*Bush et al., 2017*; *Nilsen and Graveley, 2010*).

## What is the role of U6 snRNA m$^6$A modification in 5' and 3'SS selection?

Our analysis of available cryo-EM structures of human spliceosomes (*Bertram et al., 2017*; *Bertram et al., 2020*; *Fica et al., 2019*; *Zhang et al., 2018*) does not provide clear evidence of a direct interaction between U6 snRNA m$^6$A and a spliceosomal protein (*Figure 8B*; *Figure 8—figure supplement 1*). Instead, during the transfer of the 5'SS from U1 snRNA to U6 and U5 snRNAs in spliceosomal B complexes, U6 AC$^{m6}$**A**GA m$^6$A faces 5'SS A$_{+4}$ in a trans Hoogsteen sugar edge interaction that could stabilise the 5'SS/U6 helix by capping (*Figure 8B*). This is important because 5'SS A$_{+3}$ is not aligned to stack on A$_{+4}$ at this stage. However, later in spliceosomal B$^{act}$ and C complexes, 5'SS A$_{+3}$ engages in a more stable interaction with the U6 snRNA G44 adjacent to m$^6$A (AC$^{m6}$A$\underline{GA}$), stacking on top of the 5'SS/U6 helix and stabilising it (*Figure 8C*; *Figure 8—figure supplement 1*). Such stabilisation is important because degenerate 5'SS sequences mean that the helix formed with U6 snRNA AC**A**GA is relatively weak, short, and comprised mostly of non-canonical RNA-RNA interactions (*Figure 8B*).

Our global RNA-seq analyses reveal that m$^6$A-modified U6 snRNA pairs less readily with 5'SSs containing U$_{+4}$ than those containing A$_{+4}$. Biophysical data from model RNAs provide a possible explanation for this finding: an m$^6$A-U base pair can form in a duplex, but the methylamino group rotates from a *syn* geometry on the Watson–Crick face to a higher-energy anti-conformation, positioning the methyl group in the major groove (*Roost et al., 2015*). As a result, m$^6$A has a destabilising effect on A−U basepairs in short RNA helices (*Kierzek and Kierzek, 2003*; *Roost et al., 2015*). Therefore, 5'SS U$_{+4}$ may be selected less frequently by m$^6$A-modified U6 snRNA because of destabilisation of an already weak helix. In contrast, the thermal stability of m$^6$A:A is increased compared to A:A (*Roost et al., 2015*). Biophysical measurements also indicate m$^6$A in unpaired positions base stacks more strongly than the unmodified base, adding substantial stabilisation to adjacent duplexes (*Roost et al., 2015*). In the absence of U6 snRNA m$^6$A modification, our data reveal that 5'SS U$_{+4}$ sites are preferred. This is consistent with a relatively strong interaction being important at the terminal position of the U6 snRNA ACAGA helix in the B complex to stabilise its formation.

The potential role of U6 snRNA m⁶A in 3'SS selection may be explained by U6 AC**A**GA interactions during the remodelling of the spliceosome for the second splicing reaction (*Figure 8C*; *Figure 8— figure supplement 1*). In humans, RNA rearrangements during the C to C\* transition include stacking of the 3'SS $G_{-1}$ onto U6 A45 (AC**A**G$\underline{A}$), which remains paired to $U_{+2}$ of the 5'SS (*Fica, 2020*; *Fica et al., 2019*; *Wilkinson et al., 2017*). This results in a continuous helix stack, involving the interaction between U6 A43 and 5'SS $A_{+4}$, which forms the receptor onto which the 3'SS docks. Our data are consistent with a model in which a strong interaction between U6 and the 5'SS +4 position stabilises this receptor, enabling 3'SSs distal to the branchpoint to compete more efficiently with proximal 3'SSs that are favoured by scanning (*Smith et al., 1993*). A more stable receptor could allow the ATPase Prp22, which proofreads exon ligation (*Mayas et al., 2006*), to promote more efficient sampling and usage of distal 3'SSs. This has previously been observed in *S. cerevisiae* (*Semlow et al., 2016*), where the U6/5'SS helix is intrinsically stronger than in *Arabidopsis* and humans (*Plaschka et al., 2019*; *Wilkinson et al., 2020*).

## A regulatory or adaptive role for U6 snRNA m⁶A modification?

Alternative splicing is the result of competition between multiple splicing choices. Our analysis of annotated alternative 5'SS pairs in *Arabidopsis* (and humans) revealed that when one of the alternative 5'SS pair exhibited strong U5 snRNA loop 1 recognition, the other was more likely to have stronger complementarity to U6 snRNA AC**A**GA box. This suggests that alternative splicing could be regulated by changing the relative favourability of U5 and U6 snRNA interactions.

It is an open question as to whether m⁶A modification of U6 snRNA is regulated. Either the activity of FIO1 might be controlled or demethylases might act directly upon U6 snRNA. There is a precedent for the control of splicing by regulation of U6 snRNA modifications because pseudouridylation of *S. cerevisiae* U6 snRNA, which controls entry into filamentous growth, can increase the splicing efficiency of suboptimal introns (*Basak and Query, 2014*). Human UsnRNAs are targeted by demethylases. For example, FTO targets cap adjacent methylation (m⁶Am) in UsnRNAs (except U6), and this activity may account for splicing differences detected in FTO knockout backgrounds (*Mauer et al., 2019*). Notably, *Arabidopsis* mutants defective in the function of the RNA demethylase ALKBH10b flower late (*Duan et al., 2017*).

Two genome-wide association studies have identified sequence differences at or near the *FIO1* locus associated with natural variation in *Arabidopsis* ecotypes' flowering time (*Price et al., 2020*; *Sasaki et al., 2015*). It will be important to test whether genetic variation alters the efficiency of U6 snRNA m⁶A modification by FIO1 in different ecotypes and what impact this has upon global splicing patterns, including for the regulators of flowering time that we characterise here. There are 13 genes encoding U6 snRNA in the *Arabidopsis* Col-0 genome (Wang and Brendel, 2004). Currently, we know little about the relative patterns of U6 snRNA gene expression or m⁶A modification status and how this may vary in different ecotypes. Indeed, this paucity of knowledge on U6 snRNA variants' modification and expression applies to humans too. Our study on U6 snRNA m⁶A modification has important implications for understanding the mechanism of splicing and the evolution of alternative splicing. It will now be interesting to investigate the possibility that modulation of U6 snRNA m⁶A may be regulatory or adaptive.

## Materials and methods
### Plant material
The wild-type Col-0 accession, *fio1-3* (SALK_084201), and *maf2* (SALK_045623) were obtained from Nottingham *Arabidopsis* Stock Centre. The *fio1-1* mutant was a gift from Prof Hong Gil Nam (Daegu Gyeongbuk Institute of Science and Technology), Republic of Korea. The *fip37-4* mutant was a gift from Prof Rupert Fray (University of Nottingham), UK.

### Plant growth conditions
Seeds of wild-type Col-0, *fip37-4*, and *fio1-1* used for nanopore DRS and m⁶A immunopurification were surface sterilised and sown on MS10 media plates supplemented with 2% agar, stratified at 4°C for 2 days, germinated in a controlled environment at 20°C under 16 hr light/8 hr dark conditions, and harvested 14 days after transfer to 20°C. Seeds of wild-type Col-0 and *fio1-3* used for Illumina

RNA-seq were surface sterilised and sown on 0.5 MS media plates supplemented with 2% agar, stratified at 4°C for two days, germinated in a controlled environment at 20°C under 16 hr light/8 hr dark conditions for 8 days. Seedlings were then transferred to either 28, 20, 12, or 4 °C for 4 hr under dark conditions before harvesting. Seeds used for *MAF2* splicing analysis were sown on 0.5 MS media plates supplemented with 2% agar, stratified at 4°C for two days, germinated in a controlled environment at 20°C under 16 hr light/8 hr dark conditions, and harvested 14 days after transfer to 20°C.

## Mutant screen

The two-step EMS mutant screen was conducted in a Col-0 line carrying a homozygous transgene in which the genomic *MAF2* coding region was translationally fused to luciferase (*gMAF2:LUC*). A gateway cloning approach was used to introduce *gMAF2:LUC* into the Alligator vector pFP101, which features a recombination site downstream of a CaMV 35 S promoter and selection by Green Fluorescent Protein (GFP) fluorescence (*Bensmihen et al., 2004*). *Agrobacterium tumefaciens* strain GV3101 was used to transform Col-0 via the floral dip method (*Clough and Bent, 1998*). A homozygous line with strong LUC expression and a clear GFP fluorescence in the seed coat (Alligator selective marker) was identified and used for EMS mutagenesis.

Approximately, 20,000 M0 seeds of the *gMAF2:LUC* line were soaked overnight in 100 mL of phosphate buffer at 4°C. The buffer was replaced, with EMS added to a final concentration of 25 mM, and the seeds were incubated at room temperature with gentle agitation for 16 hr. The EMS was neutralised with 1 M NaOH, and the seeds were gently washed twice with 100 mM sodium thiosulphate for 15 min, followed by three 15 min washes in distilled water. The M1 seeds were air-dried on filter paper overnight, before planting on soil. The resulting plants were allowed to self-pollinate and grow to maturity. M2 seeds were collected from all plants. Approximately, 15,000 seeds were screened in two steps. In step 1, stratified seeds were planted on soil and grown in controlled environment chambers at 16°C, 16 hr light/8 hr dark, with light intensity of 200 µmol/m$^2$/s. The first 100 plants to flower were selected for further screening. Splicing of *MAF2* intron 3 was assessed by RT-qPCR to identify early flowering lines showing enhanced intron retention at 16°C. Although the parental line had the *gMAF2:LUC* reporter, luciferase activity was not monitored at any stage. Leaf material was frozen and homogenized using QIAGEN TissueLyser LT. RNA was extracted using a Nucleospin II RNA extraction kit (Machery–Nagel). Total RNA (1.5 µg per sample) was reverse-transcribed using the High Capacity cDNA Reverse Transcription Kit with Rnase Inhibitor (Applied Biosystems). Amplification of *MAF2* was performed according to manufacturer's guidelines for 35 PCR cycles using primers 3 and 4 from a previous study (*Rosloski et al., 2013*). These primers, which span *MAF2* intron 3, generate two products, which correspond to *MAF2 var2* (*MAF2* intron 3 retained) and *MAF2 var1* (*MAF2* intron 3 excised), respectively. PCR amplified products were separated on a 2% w/v agarose gel to resolve the splice variants. M2 plants showing a significantly enhanced *var2:var1* ratio compared to the parental line control were selected for further testing.

## Bulk segregant analysis for mutant mapping

The EMS-129 line was backcrossed to the parental *gMAF2:LUC* reporter line to generate an F2 segregating population. Phenotypically early flowering and wild-type flowering time plants were identified from this F2 population and separated for bulk DNA sequencing. Genomic DNA was extracted using a Qiagen Plant DNA Maxi kit. Sequencing was carried out by the University of Leeds Next Generation Sequencing (NGS) facility using a HiSeq3000 sequencer with a 150 bp paired end library.

Sequenced reads were mapped to the TAIR10 reference sequence using bwa-mem2 (*Vasimuddin et al., 2019*). SNPs were called using bcftools and filtered for EMS-induced G>A transitions only (*Danecek et al., 2021*). Functional consequences of each SNP were predicted using SnpEff (*Cingolani et al., 2012*). Allele fraction plots were generated from vcf files using matplotlib. Significance testing was performed for each SNP using G-tests and smoothed using a tri-cube kernel with a window size of 2 megabases.

## Nanopore DRS
### Total RNA isolation
Total RNA was isolated using Rneasy Plant Mini kit (Qiagen) and treated with TURBO Dnase (ThermoFisher Scientific). The total RNA concentration was measured using a Qubit 1.0 Fluorometer and

Qubit RNA BR Assay Kit (ThermoFisher Scientific). RNA quality and integrity were assessed using the NanoDrop 2000 spectrophotometer (ThermoFisher Scientific) and Agilent 2200 TapeStation System (Agilent).

## Preparation of libraries for direct RNA sequencing of poly(A)+ mRNA using nanopores

Total RNA was isolated from the Col-0, *fip37-4*, and *fio1-1* seedlings as detailed above. mRNA isolation and preparation of nanopore DRS libraries using the SQK-RNA002 nanopore DRS Kit (Oxford Nanopore Technologies) were performed as previously described (*Parker et al., 2020*). Libraries were loaded onto R9.4 SpotON Flow Cells (Oxford Nanopore Technologies) and sequenced using a minION device for a 48 hr runtime. Four biological replicates were performed for each genotype.

### Nanopore DRS mapping

Nanopore DRS reads were basecalled using Guppy version 3.6.0 high accuracy RNA model. For mRNA modification analysis, reads were mapped to the Araport11 reference transcriptome (*Cheng et al., 2017*) using minimap2 version 2.17 (*Li, 2018*), with parameters -a -L –cs =short k14 –for-only –secondary =no. For other analyses, reads were mapped to the *Arabidopsis TAIR10* genome (*Arabidopsis Genome Initiative, 2000*) using two pass alignment with minimap2 version 2.17 and 2passtools version 0.3 (*Parker et al., 2021b*). First pass minimap2 alignment was performed using the parameters -a -L –cs =short -x splice -G20000 –end-seed-pen 12 -uf. 2passtools score was then run with default parameters on each replicate to generate junctions, followed by 2passtools merge to combine them into a final set of guide junctions. Reads were remapped with minimap2 using the same parameters but with the addition of the guide junctions using –junc-bed and –juncbonus =10. Pipelines for processing of nanopore DRS data were built and executed using Snakemake version 6.15.3 (*Köster and Rahmann, 2012*). Public nanopore DRS data downloaded from ENA accessions PRJNA749003 (*Xu et al., 2022*) and PRJNA877932 (*Sun et al., 2022*) were processed in the same way as newly generated data.

### Nanopore poly(A)+ mRNA modification analysis

Differential modification analysis was performed on Col-0, *fip37-4*, and *fio1-1* data using the 'n-sample' GitHub branch of Yanocomp (*Parker et al., 2021a*). Kmer-level signal data were generated using f5c eventalign version 0.13.2 (*Gamaarachchi et al., 2020*; *Loman et al., 2015*) and Yanocomp prep. A three-way comparison between the genotypes was performed using Yanocomp gmmtest, with a minimum KS statistic of 0.25. A 5% false discovery rate threshold was used to identify transcriptomic sites with significant changes in modification rate. Motif analysis was performed using meme version 5.1.1 with the parameters -cons NNANN -minw 5 -maxw 5 -mod oops (*Bailey et al., 2015*). Differential poly(A) site usage was performed using d3pendr version 0.1 with default parameters and thresholded using a 5% false discovery rate and an effect size (measured using earth mover distance) of 25 (*Parker et al., 2021c*).

## Illumina RNA sequencing

### Preparation of libraries for Illumina RNA sequencing

Total RNA was isolated from Col-0 and *fio1-3* seedlings using Nucleospin RNA kit (Macherey–Nagel, 740955) and treated with rDNase (Macherey–Nagel, CAS 9003-98-9) on columns according to manufacturers' instructions. RNA concentration, quality, and integrity were assessed using the NanoDrop 1000 spectrophotometer (Labtech) and agarose gel electrophoresis. Poly(A)+RNA purification and Illumina RNA-seq library preparation were performed by Genewiz UK Ltd. Poly(A)+RNA was selected with NEBNext Poly(A) mRNA Magnetic Isolation Module. Preparation of the sequencing libraries was performed using the NEBNext Ultra II Directional RNA Library Prep Kit for Illumina (New England Biolabs). 150 bp paired-end sequencing was carried out using Illumina Novaseq 6000. Six biological replicates were performed for each genotype.

## Illumina RNA sequencing data processing

Illumina RNA-seq data were assessed for quality using FastQC version 0.11.9 and MultiQC version 1.8 (*Andrews, 2017*; *Ewels et al., 2016*). Reads were mapped to the TAIR10 genome using STAR version 2.7.3 a (*Dobin et al., 2013*) with a splice junction database generated from the Araport11 reference annotation (*Cheng et al., 2017*). We used Stringtie version 2.1.7 in mix mode (*Kovaka et al., 2019*) to generate condition-specific transcriptome assemblies from Illumina RNA-seq replicates and pooled Nanopore DRS data. All assemblies were merged with the Araport11 reference annotation using Stringtie merge to create a unified set of transcripts for quantification. Transcript open reading frames were annotated using Transuite version 0.2.2 (*Entizne et al., 2020*), which was also used to predict nonsense-mediated decay sensitivity. Pipelines were written and executed using Snakemake version 6.15.3 (*Köster and Rahmann, 2012*).

Transcripts were quantified using Salmon version 1.1.0 (*Patro et al., 2017*) with the TAIR10 genome assembly as decoys. SUPPA version 2.3 was used to generate event level PSIs from transcript level quantifications (*Trincado et al., 2018*). PSIs were loaded into Python 3.6.7 using pandas version 1.0, and generalised linear models (GLMs) were fitted per event using statsmodels version 0.11 (*Harris et al., 2020*; *McKinney, 2010*; *Oliphant, 2007*; *Seabold and Perktold, 2010*). GLMs were used to test the relationship of PSI with genotype, temperature, and genotype×temperature interaction. Calculated p values were adjusted for multiple testing using the Benjamini–Hochberg false discovery rate method. A false discovery rate of 5% was chosen to threshold events with significant changes in PSI which correlated with genotype, temperature, or genotype×temperature. Changes in motif composition were tested using G-tests of base frequencies at the −2 to +5 position of the splice site. To generate sequence logos, 5′SS or 3′SS from sets of alternative splicing events was filtered to remove duplicated positions, and probability logos were plotted using matplotlib version 3.3 and matplotlib_logo (*Hunter, 2007*; *Parker, 2022b*). Contingency tables of splice site classes at U5 and U6 interacting positions were generated using the difference of the −2 to −1 positions of the 5′SS from the consensus motif AG, and the difference of the +3 to +5 positions of the 5′SS from the consensus motif RAG, respectively. Classes were ordered by their log-likelihoods using PSSMs generated from all annotated 5′SSs (described below). Heatmaps of contingency tables were generated with seaborn version 0.11 (*Waskom, 2021*). Gene tracks using reads aligned to the TAIR10 reference genome were generated using pyBigWig version 0.3.17, pysam version 0.18, and matplotlib version 3.3 (*Heger et al., 2014*; *Hunter, 2007*; *Ramírez et al., 2014*).

Public Illumina RNA-seq datasets were downloaded from GSA accession CRA004052 (*Wang et al., 2022*) and ENA accession PRJNA877932 (*Sun et al., 2022*). Quality control was performed using FastQC and MultiQC as described above (*Andrews, 2017*; *Ewels et al., 2016*). Col-0 replicate 1 of dataset CRA004052 was discarded due to extreme GC-bias, likely caused by PCR overamplification and/or low RNA input. Transcript-level quantification and PSI estimation were performed as described above. PSIs were used to estimate splicing changes in *fio1* mutant alleles using SUPPA2 diffSplice (*Trincado et al., 2018*).

## Global splice site analyses

To measure the predicted strengths of U5 snRNA loop 1 and U6 snRNA ACAGA box interactions for individual sequences, all 5′SSs annotated in the Araport11 reference transcriptome (*Cheng et al., 2017*) were used to generate log transformed PSSMs. The U5 and U6 log-likelihood scores for individual sequences were then calculated using the −2 to −1 and +3 to +5 positions, respectively. Correlation of U5 and U6 PSSM scores was calculated using Spearman rank correlation coefficient. This analysis was repeated using 5′SSs from the *H. sapiens* GRCh38 (*International Human Genome Sequencing Consortium, 2004*), *C. elegans* Wbcel235 (*The C. elegans Sequencing Consortium, 1998*), *D. melanogaster* BDGP6 (*dos Santos et al., 2015*), and *D. rerio* GRCz11/danRer11 (*Howe et al., 2013*) genome assemblies and annotations, downloaded from Ensembl Genomes release 104 (*Howe et al., 2021*). To measure relative U5 and U6 interactions for pairs of alternative 5′SSs, SUPPA was used to identify pairs of alternative 5′SSs in each genome annotation (*Trincado et al., 2018*). These alternative 5′SS pairs were ordered by their genomic positions relative to the strand of the parent gene (i.e. upstream and downstream), and log-odds ratios were calculated using downstream 5′SS log-likelihoods as the denominator. Correlation of U5 and U6 log-odds ratios was calculated using Spearman rank correlation coefficient. Scatter plots with linear regression lines were plotted

using seaborn (**Waskom, 2021**), and 95% CIs for regression lines were calculated using bootstrap sampling with replacement.

## Immunopurification and detection of m⁶A modified RNA

### Synaptic systems anti-m⁶A

Total RNA was purified from ~300 mg of frozen plant tissue using the miRVana miRNA isolation kit (Ambion) and treated with DNase I (New England Biolabs) according to the manufacturer's instructions. The quantity and integrity of RNA were checked using a NanoDrop 2000 spectrophotometer (Thermo Fisher Scientific) and Agilent 2200 TapeStation System. Approximately 5 µg of RNA was suspended in 500 µL low-salt buffer (50 mM Tris-HCl pH 7.4, 150 mM NaCl, and 0.5% v/v NP-40), with 80 U RNAsin Plus RNase inhibitor, (Promega) and 10 µL (1 mg/mL) m⁶A-rabbit polyclonal purified antibody (#202 003 Synaptic Systems). The samples were mixed by rotation for 2 hr at 4°C. Protein A/G magnetic beads (Pierce, ThermoFisher) were washed twice with low-salt buffer, then added to each RNA/antibody sample and mixed with rotation at 4°C for 16 hr. The beads were washed twice with high-salt buffer (50 mM Tris-HCl pH 7.4, 0.5 M NaCl, and 1% v/v NP-40), twice with low-salt buffer, and twice with Polynucleotide kinase (PNK) wash buffer (20 mM Tris-HCl pH 7.4, 10 mM MgCl₂, and 0.2% v/v Tween-20). Immunopurified RNA was eluted by digestion with proteinase K in 200 µL of proteinase K (PK) buffer (50 mM Tris-HCl pH 7.4, 150 mM NaCl, 1 mM EDTA, and 0.1% w/v SDS) at 37°C for 20 min with shaking at 300 rpm, followed by phenol-chloroform extraction and sodium acetate/ethanol precipitation.

Immunopurifed RNA (45 ng) was reverse transcribed (RT) with U6 and U2 snRNA reverse primers using SuperScript III Reverse Transcriptase (ThermoFisher), according to the manufacturer's instructions. RT-qPCR was carried out using the SYBR Green I (Qiagen) mix with primers targeted to U2 and U6 snRNA (**Supplementary file 2**). The specificity of RT-qPCR was confirmed by sequencing of amplified products.

### Millipore anti-m⁶A

Total RNA was purified from ~600 mg of frozen plant tissue using the miRVana miRNA isolation kit (Ambion) and treated with DNase I (New England Biolabs), according to the manufacturer's instructions. The quantity and integrity of RNA were checked using a NanoDrop 1000 spectrophotometer (Thermo Fisher Scientific). After sodium acetate/ethanol precipitation, approximately 10 µg of RNA was resuspended in 20 µL nuclease free water, with 5% of the RNA being kept for the input sample.

Protein A/G magnetic beads (Millipore, 16–663) were washed twice with m⁶A wash buffer (10 mM TrisHCl pH7.4, 150 mM NaCl, and 0.1% v/v NP-40) and then resuspended in 50 µL m⁶A wash buffer. 5 µL (1 mg/mL) of Anti-N6-methyladenosine (m⁶A) antibody (Millipore, ABE572) or Rabbit Anti-IgG (Thermofisher, A16104) was coupled to the washed beads on a roller for 40 min at room temperature. After incubation, beads were washed three times in m⁶A wash buffer. m⁶A immunoprecipitation buffer was then added to the beads (10 mM TrisHcl pH7.4, 150 mM NaCl, 0.1% v/v NP-40, and 17.5 mM EDTA pH 8) with 80 U SUPERase•In RNase Inhibitor, and 10 µL of each RNA sample was added to both the anti-m⁶A and anti-IgG antibody/beads mixture. Samples were mixed by rotation at 4°C for 16 hr. The beads were washed five times with m⁶A wash buffer. Immunopurified and 5% input RNA was eluted by digestion with proteinase K in 150 µL of PK buffer (50 mM Tris-HCl pH 7.4, 150 mM NaCl, 0.1% v/v NP-40, and 0.1% w/v SDS) at 55°C for 30 min, followed by extraction with TRIzol LS (10296028, Thermofisher) and sodium acetate/ethanol precipitation.

Immunopurifed RNA (500 ng) was RT with random hexamers using ThermoFisher MultiscribeTM II Reverse Transcriptase (4311235, Thermo Fisher Scientific) according to the manufacturer's instructions. RT-qPCR was carried out using the SYBR Green I (Qiagen) mix with primers targeted to U2 and U6 snRNA (**Supplementary file 2**). The specificity of RT-qPCR was confirmed by sequencing of amplified products.

### m⁶A liquid chromatography with tandem mass spectrometry

Total RNA was isolated as described above. Poly(A)+RNA was purified using the Dynabeads mRNA purification kit (Thermo Fisher Scientific) according to the manufacturer's instructions. The quality and quantity of mRNA were assessed using a NanoDrop 2000 spectrophotometer and Agilent 2200 TapeStation System. Samples for m⁶A LC-MS/MS were prepared as previously described (**Huang**

*et al., 2018*) with modifications according to *Parker et al., 2021c*. LC-MS/MS was carried out by the FingerPrints Proteomics facility at the University of Dundee. m⁶A/A ratio quantification was performed in comparison with the curves obtained from pure adenosine (endogenous P1 receptor agonist, Abcam) and m⁶A (modified adenosine analog, Abcam) nucleoside standards. Statistical analysis was performed using a two-way *t*-test.

## Code availability

All pipelines, scripts, and notebooks used to generate figures are available from GitHub at https://github.com/bartongroup/Simpson_Davies_Barton_U6_methylation and Zenodo at https://zenodo.org/record/6372644 (*Parker, 2022a*).

## Acknowledgements

We thank Mary McKay (School of Biology, University of Leeds) for excellent technical assistance with the EMS mutant screen. We are indebted to Oliver Manners for help in establishing m⁶A immuno-precipitation. We thank Abdelmadjid Atrih (Centre for Advanced Scientific Technologies) for the m⁶A LC-MS/MS analysis. We thank David Lilley, Tim Wilson and Adrian Whitehouse for helpful discussions. We thank Alper Akay, Martin Balcerowicz, James Lloyd, Anjil Srivastava and Carey Metheringham for comments on the manuscript. This work was supported by awards from the BBSRC (BB/V010662/1; BB/M010066/1; BB/M004155/1; BB/W007673/1) and the University of Dundee Global Challenges Research Fund to GGS and GJB; and awards from BBSRC (BB/M000338/1; BB/W007967/1) and ERA-CAPS FLOWPLAST to BD. BS is funded through the BBSRC DTP3 award, BB/T007222/1. KK and NJ were funded through the European Union Horizon 2020 research and innovation programme under Marie Skłodowska-Curie grant agreements 799300 and 896598, respectively. SMF is a Wellcome Trust and Royal Society Sir Henry Dale Fellow (grant number 220212/Z/20/Z). The FingerPrints Proteomics Facility of the University of Dundee is supported by a Wellcome Trust Technology Platform Award (097945/B/11/Z).

## Additional information

### Funding

| Funder | Grant reference number | Author |
|---|---|---|
| Biotechnology and Biological Sciences Research Council | BB/M000338/1 | Brendan H Davies |
| Biotechnology and Biological Sciences Research Council | BB/W007967/1 | Brendan H Davies |
| Biotechnology and Biological Sciences Research Council | BB/T007222/1 | Beth K Soanes |
| HORIZON EUROPE Marie Sklodowska-Curie Actions | 799300 | Katarzyna Knop |
| HORIZON EUROPE Marie Sklodowska-Curie Actions | 896598 | Nisha Joy |
| Wellcome Trust | 220212/Z/20/Z | Sebastian M Fica |
| Global Challenges Research Fund | University of Dundee Global Challenges Research Fund | Geoffrey J Barton Gordon G Simpson |
| Biotechnology and Biological Sciences Research Council | BB/W007673/1 | Gordon G Simpson |

| Funder | Grant reference number | Author |
|---|---|---|
| Biotechnology and Biological Sciences Research Council | BB/V010662/1 | Geoffrey J Barton Gordon G Simpson |
| Biotechnology and Biological Sciences Research Council | BB/M010066/1 | Geoffrey J Barton Gordon G Simpson |
| Biotechnology and Biological Sciences Research Council | BB/M004155/1 | Geoffrey J Barton Gordon G Simpson |

The funders had no role in study design, data collection and interpretation, or the decision to submit the work for publication. For the purpose of Open Access, the authors have applied a CC BY public copyright license to any Author Accepted Manuscript version arising from this submission.

## Author contributions

Matthew T Parker, Conceptualization, Data curation, Software, Formal analysis, Validation, Investigation, Visualization, Methodology, Writing – original draft, Writing – review and editing; Beth K Soanes, Conceptualization, Data curation, Software, Formal analysis, Validation, Investigation, Visualization, Methodology, Writing – original draft, Project administration, Writing – review and editing; Jelena Kusakina, Conceptualization, Formal analysis, Validation, Investigation, Methodology, Writing – original draft, Project administration, Writing – review and editing; Antoine Larrieu, Conceptualization, Resources, Formal analysis, Validation, Investigation, Methodology, Project administration, Writing – review and editing; Katarzyna Knop, Conceptualization, Resources, Formal analysis, Funding acquisition, Validation, Investigation, Methodology, Project administration, Writing – review and editing; Nisha Joy, Friedrich Breidenbach, Conceptualization, Formal analysis, Funding acquisition, Validation, Investigation, Methodology, Project administration, Writing – review and editing; Anna V Sherwood, Formal analysis, Validation, Investigation, Methodology, Project administration, Writing – review and editing; Geoffrey J Barton, Conceptualization, Resources, Supervision, Funding acquisition, Project administration, Writing – review and editing; Sebastian M Fica, Conceptualization, Formal analysis, Funding acquisition, Visualization, Writing – original draft, Project administration, Writing – review and editing; Brendan H Davies, Gordon G Simpson, Conceptualization, Resources, Supervision, Funding acquisition, Writing – original draft, Project administration, Writing – review and editing

## Author ORCIDs

Matthew T Parker http://orcid.org/0000-0002-0891-8495
Katarzyna Knop http://orcid.org/0000-0002-2636-9450
Friedrich Breidenbach http://orcid.org/0000-0002-9610-1927
Anna V Sherwood http://orcid.org/0000-0001-7153-8420
Geoffrey J Barton http://orcid.org/0000-0002-9014-5355
Gordon G Simpson http://orcid.org/0000-0001-6744-5889

## Decision letter and Author response

Decision letter https://doi.org/10.7554/eLife.78808.sa1
Author response https://doi.org/10.7554/eLife.78808.sa2

# Additional files

## Supplementary files

• Supplementary file 1. Read statistics for Nanopore direct RNA sequencing (DRS) and Illumina RNA-seq datasets.

• Supplementary file 2. Oligos and primers used in this study.

• MDAR checklist

## Data availability

Illumina sequencing data from the genetic screen that identified *fio1-4* is available from ENA accession PRJEB51468. Col-0, *fip37-4* and *fio1-1* nanopore DRS data is available from ENA accession PRJEB51364. Col-0 and *fio1-3* Illumina RNA-Seq data is available from ENA accession PRJEB51363.

The following datasets were generated:

| Author(s) | Year | Dataset title | Dataset URL | Database and Identifier |
|---|---|---|---|---|
| Parker MT, Soanes BK, Kusakina J, Larrieu A, Knop K, Joy N, Briedenbach F, Sherwood A, Barton GJ, Fica SM, Davies B, Simpson GG | 2022 | Mutant screen of early flowering Arabidopsis EMS mutants with increased MAF2 intron retention | https://www.ebi.ac.uk/ena/browser/view/PRJEB51468 | European Nucleotide Archive (EMBL-EBI), PRJEB51468 |
| Parker MT, Soanes BK, Kusakina J, Larrieu A, Knop K, Joy N, Briedenbach F, Sherwood A, Barton GJ, Fica SM, Davies B, Simpson GG | 2022 | Nanopore direct RNA sequencing of Col-0 and fio1-3 mutant Arabidopsis | https://www.ebi.ac.uk/ena/browser/view/PRJEB51364 | European Nucleotide Archive (EMBL-EBI), PRJEB51364 |
| Parker MT, Soanes BK, Kusakina J, Larrieu A, Knop K, Joy N, Briedenbach F, Sherwood A, Barton GJ, Fica SM, Davies B, Simpson GG | 2022 | Illumina RNA sequencing of Col-0 and fio1-3 mutant Arabidopsis | https://www.ebi.ac.uk/ena/browser/view/PRJEB51363 | European Nucleotide Archive (EMBL-EBI), PRJEB51363 |

The following previously published datasets were used:

| Author(s) | Year | Dataset title | Dataset URL | Database and Identifier |
|---|---|---|---|---|
| Xu T, Wu X, Wong CE, Fan S, Zhang Y, Zhang S, Liang Z, Yu H | 2022 | N6-Adenosine (m6A) Methylation profiling in *Arabidopsis thaliana* wild-type and fio1-2 by nanopore DRS sequencing | https://www.ebi.ac.uk/ena/browser/view/PRJNA749003 | European Nucleotide Archive (EMBL-EBI), PRJNA749003 |
| Sun B, Bhati KK, Song P, Edwards A, Petri L, Kruusvee V, Blaakmeer A, Dolde U, Rodrigues V, Straub D, Yang J, Jia G, Wenkel S | 2022 | Next Generation Sequencing of Wild Type (Col-0), fio1-1 and fio1-2 methylated RNA immunoprecipitation and transcriptome in *Arabidopsis thaliana* | https://www.ncbi.nlm.nih.gov/geo/query/acc.cgi?acc=GSE171928 | NCBI Gene Expression Omnibus, GSE171928 |
| Wang C, Yang J, Song P, Zhang W, Lu Q, Yu Q, Jia G | 2022 | Identification and Function Research of Methyltransferase of N6 Methyladenine in Plants | https://ngdc.cncb.ac.cn/gsa/browse/CRA004052 | Genome Sequence Archive (CNCB), CRA004052 |

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
