## [Editor Report]

This is an important paper reporting that an adenosine methyltransferase in the model plant Arabidopsis functions to target a key RNA component of the spliceosome, as in fission yeast, and thereby contributes to intron recognition as well as flowering timing. By contrast, the authors report no major role for the methyltransferase in targeting mRNAs, as reported in several previous studies in Arabidopsis. Overall, the approaches are convincing, although a correlative analysis identifying an intronic sequence feature characteristic of methyltransferase-sensitive introns is not followed up with tests to establish causality of this feature.

---

## [Decision Letter]

**Decision letter after peer review:**

Thank you for submitting your article "m^6^A modification of U6 snRNA modulates usage of two major classes of pre-mRNA 5' splice site" for consideration by *eLife*. Your article has been reviewed by 2 peer reviewers, and the evaluation has been overseen by a Reviewing Editor and James Manley as the Senior Editor. The reviewers have opted to remain anonymous.

Essential revisions:

1. A previous report suggested limited splicing changes in the fio1 mutant. The authors should add a comparison and discussion. In addition, FLC mRNA can be m6A methylated. The authors appear to suggest the effect is secondary. More analysis and explanation are required. For instance, the authors could measure m6A level on FLC in fio1 mutants and mta mutants, and compare them with that of wild tpye.

2. The authors used Nanopore m6A sequencing to map m6A in mRNA from wt and fio1 mutant strains. We would suggest either RIP-seq or mass spectrometry measurement to confirm the loss of fio1 leads to limited mRNA m6A changes.

3. p 3 and Figure 1: "Figure 1. Schematic outline of sequential steps of pre-mRNA splicing." I don't see why this is a main figure. It's the type of schematic that could be found in any splicing review. This seems like a Supplemental figure, at most. Instead, a schematic of their screen, described in the first paragraph of the Results, including some pictures of WT and early flowering Arabidopsis mutants obtained from the screen, would be more appropriate (and is needed). The authors should also show the RT-PCR data supporting enhanced MAF2 intron 3 retention at 16{degree sign}C. Overall, the authors provide few details about this screen, including how many EMS mutants were screened. How did the authors identify mutants that alter MAF1 intron 3 splicing, when the difference in intron 3 retention (as shown in Figure 4C) is so subtle?

4. p 6. "Crosses between EMS 129 and either fio1-1, or a Transfer-DNA insertion line disrupting AT2G21070 (SALK_084201; fio1-3) confirmed allelism." No data for this are shown; data supporting these conclusions should be included.

5. Many of the figures are difficult to interpret. For example, Figure 3 legend gives some details of the bioinformatics, but does not say anything about what the actual experiments were. The text is overly-focused on bioinformatics without an adequate description of the experiments that were performed. Please provide information detailing the actual experiments; without this information, these parts of the manuscript are both uninformative and impossible to evaluate.

6. Figure 4C. The change in MAF2 intron 3 retention is seemingly tiny. What exactly is the deltaPsi for this? Can this be confirmed by RT-PCR or RT-qPCR? This highlights an issue raised above: how did the authors identify mutants in the screen by RT-PCR that altered MAF2 intron 3 retention if the change is this subtle?

7. I found the description of the SS sequence enrichment incredibly long (7 pages of text) and overwhelmingly descriptive without a view of what the conclusions would be. Sequence motifs in the context of otherwise disparate intron and exon sequences can be suggestive of a mechanism; but, in the end, confirmation of these sequence effects requires experimental testing within a constant gene sequence. This is necessary to make any solid conclusion based on rigorous science. These are critical experiments that are missing in this manuscript.

8. The model seems to be that U5 and U6 RNAs contribute to a platform for recognition of 5'SSs: a 5'SS must interact sufficiently with this platform in order to make it through splicing; stronger interaction as some positions can compensate for weaker interactions at other positions. I don't think that there's anything new here in terms of mechanism. m6A in U6 contributes to stability; this was already postulated in the *S. pombe* system (Ishigami et al. 2021). Is this biologically important? Does FIO1 expression change developmentally? or, is it a constitutive feature of U6-5'SS interaction? These questions are not addressed or even raised in this manuscript.

9. p 18. The authors state that "FIO1 buffers spicing fidelity" and "that FIO1 function calibrates the temperature range over which MAF2 alternative slicing occurs", but I don't see any data that directly support these assertions. These seem like extrapolations of the current data that require experimental tests to support them.

10. The authors refer to the "cooperative roles for U5 and U6 snRNA in splice site selection" in the abstract. I don't see any evidence for interactions with U5 in this manuscript, and the proposed role of U6 is not tested experimentally. These two deficits are the most disappointing aspects of this work.

Additional comments:

1. The main text goes on for 31 pages, >12,000 words (not including Figure Legends or References). While there are no strict limits on the length of Research Articles, the *eLife* Author Guide suggests to "try not to exceed 5,000 words in the main text". The text could be shortened significantly; conciseness would greatly benefit the readability of the manuscript.

The Discussion, at >7 pages, is excessively long and does not critically evaluate the data or conclusion presented in this manuscript.

2. p 7 and Figure 2B. What is the experimental basis of Figure 2B? The authors do not state the experimental data: is it RNA-seq or RT-PCR? In either case, some examples of the primary data for this should be shown.

3. p 6, second paragraph of the Results: "We used bulked segregant analysis and the software tool artMap (Javorka et al., 2019) to identify the causative mutation…". The authors should explain this more explicitly, for non-specialist readers. There are a number of ways in which this could have been done, but the authors do not specify what exactly was the data that led to the determination of the mutation.

4. Figure 4B. This is a comparison of fio1-3 with Col-0. I would like to see volcano plots for all the five splicing events with log2FC on the x-axis and enrichment/significance on the y axis. These bar charts with percentages are too confusing to depict what the authors are actually trying to convey and do not depict the spread of the primary data.

5. Figure 4E shows "absence of change" in MAF2 intron 5, but this intron is clearly efficiently spliced, so the comparison to inefficiently-spliced intron 3 is not particularly meaningful. A more meaningful comparison would be to a different intron that is also inefficiently spliced.

6. Figure 5. Seq logos are shown for 5'SSs that are sensitive to or that are used better in fio1-1 mutants. These can only be meaningful if they are compared to the seq logo of the overall 5'SS in A.t., which is not shown or discussed.

7. Figure 5. Why only show out to pos -2 in the exon? More positions than that can pair with U5.

8. Some of the fonts in the figures are so small as to be unreadable or nearly unreadable, e.g. Figure 4-Figure sup 1 and Figure 5-Figure sup 1.

9. ALKBH10B is a demethylase that affects early flower phenotype. Does this protein also mediate the demethylation of U6 m6A?

---

## [Author Response]

Essential revisions:1. A previous report suggested limited splicing changes in the fio1 mutant. The authors should add a comparison and discussion.

We are grateful for this suggestion. We have completed a re-analysis of recently published data that confirms the effect on splicing that we report here. We have added a new sub-section in the Results that incorporates this comparison and added new content to the Discussion that addresses this issue. We have summarised the analysis we performed below, as well as attempting to provide some additional insight into why other research groups overlooked splicing changes in their RNA-seq data.

A major finding from our study is that FIO1-dependent modification of U6 snRNA plays a significant role in defining splice site usage. This directly contradicts two recently published studies, referred to by the reviewer (Wang et al., 2022; Xu et al., 2022). Wang et al., 2022 claims that “disruption of FIONA1 does not affect global alternative splicing” and Xu et al., 2022 states “FIO1-mediated m^6^A methylation … has a limited effect on alternative splicing”. We show that these conclusions are wrong and that FIO1-induced methylation of U6 snRNA influences global splice site selection.

The Wang et al. study used two replicates per condition of Illumina RNA-seq, with ~35-45 million reads per replicate. Xu et al. used three replicates per condition of nanopore direct RNA sequencing, with ~0.5-1.2 million reads per replicate. Our study used 6 replicates per condition with four temperature treatments per genotype, for a total of 24 replicates per genotype, and a minimum of 50M pairs of 150bp Illumina RNA-seq reads per replicate (3.6 billion mapped reads in total). Consequently, our study has much greater statistical power to detect splicing changes in *fio1* mutants, compared to the previous two reports. For the purpose of this response, we have now performed a re-analysis of the previously published data, to explain the basis of the different conclusions reached in these studies.

Wang et al. performed splicing analysis using two replicates per genotype of 150bp paired-end Illumina RNA-seq, with ~35-45 million reads per replicate and used the software tool rMATs (Shen et al., 2014) to perform alternative splicing analysis. rMATs requires a transcriptome annotation file to guide splice site quantification. In contrast to our approach, Wang et al., did not generate a condition-specific transcriptome assembly for alternative splicing quantification, and presumably used the TAIR10 or Araport11 reference annotation (this is not specified in the methods section of their paper). This is important because these annotations would not include the majority of the novel alternative 5’ splice sites and intron retention events that we identified in the *fio1* mutant, and so these would not be considered in the analysis. For example, we discover ~70% fewer significant alternative splicing events when we perform splicing analysis on our own data using the reference annotation, rather than the condition-specific assembly that we generated (which includes novel splice isoforms used in *fio1* mutants) (Author response image 1 and Figure 3B). Consequently, the impact of FIO1 on splicing is likely to be underreported in the Wang et al. study.

**Author response image 1. sa2fig1:** Barplots showing the proportion of splicing events of each class, as labelled by SUPPA, which have significantly different usage (FDR < 0.05) in the *fio1-3* mutant, that are discovered when performing differential splicing analysis using the Araport11 reference annotation.

We downloaded the data from Wang et al. and performed splicing analysis using our condition-specific transcriptome annotation. We found that the Wang et al. data were of poor quality because only ~38% of reads were mappable to the Arabidopsis transcriptome. In particular, the first WT Col-0 replicate was unusable and had to be discarded because of PCR amplification bias, which can be seen in the extreme GC-content bias of sequenced reads (Figure 6—figure supplement 2A). However, we were still able to identify 384 alternative 5’SS usage events from the dataset using the software tool SUPPA2 (Trincado et al., 2018), from which we found 58% of *fio1-*sensitive 5’SSs had an A at the +4 position, compared to only 36% of 5’SSs with increased usage in the *fio1* mutant (Figure 6—figure supplement 2B, Chi2 p =3×10^-9^). We conclude that the reported limited splicing changes in the Wang et al. publication result from an underpowered dataset compounded by methodological choices in analysis.

Xu et al. performed splicing analysis using three replicates per genotype of nanopore direct RNA sequencing and used the software tool FLAIR (Tang et al., 2020) to identify differentially spliced transcripts. In our manuscript, we chose not to use nanopore direct RNA sequencing to analyse splicing because the low number of reads mean it is underpowered for splicing analysis and because the relatively high sequence error rates make the identification of splice junctions more complex (Parker et al., 2021; Tang et al., 2020). This is particularly problematic in the *fio1* mutant where cryptic alternative 5’SSs are mostly only a very short distance from the canonical 5’SS (see Page 8 and Figure 4 —figure supplement 1A in the revised manuscript). Tools like FLAIR perform post-alignment “correction” of unannotated splice junctions to nearby annotated splice junctions. This would incorrectly assign mRNAs originating from the cryptic splice sites used in the *fio1* mutant to the reference splice junction and may partly explain why alternative splicing was overlooked in this study.

We were previously able to validate splicing examples detected in our Illumina RNA-seq dataset with our own nanopore direct RNA sequencing data (Figure 4E, 5D, 6F in the revised manuscript). To confirm whether these splicing changes are also detectable in the data from Xu et al., we downloaded their basecalled data and realigned it using minimap2. We found that we could indeed identify the splicing defects we previously reported in our submitted manuscript, such as alternative 5’SS selection in *AtSAR1* intron 21 (Author response image 2), and intron retention in *WNK1* intron 5 (Author response image 2). We conclude that, like the Wang et al. study, the reported limited splicing changes in the Xu et al. publication results from an underpowered dataset compounded by methodological choices.

**Author response image 2. sa2fig2:** (A) Gene track showing alternative 5’SS selection at *AtSAR1* intron 21 in *fio1-2*, identified using nanopore DRS read alignments reanalysed from Xu et al.2022. (B) Gene track showing intron retention at *WNK1* intron 5 in *fio1-2*, identified using nanopore DRS read alignments reanalysed from Xu et al. 2022.

Finally, another recent study, by Sun et al., sequenced two replicates per condition of Illumina RNAseq derived from *fio1-1* and *fio1-5* mutants, with ~20-26 million reads per replicate (Sun et al., 2022). Sun et al. did not perform any splicing analysis on their data. We downloaded and reanalysed the data from Sun et al. using the same approach described above for the Illumina RNA-seq data from Wang et al. 2022. Although there were only two replicates, with a low sequencing depth per replicate, we found that these data were of a higher quality than the Wang et al. dataset, with a much higher mappability rate (~95%). We were able to use these data to identify 923 alternative 5’SS events caused by the *fio1-1* mutation, and 947 alternative 5’SS events caused by the *fio1-5* mutation. There was a clear preference for A_+4_ in *fio1*-sensitive 5’SSs, and for U_+4_ in 5’SSs with increased use in the *fio1* mutants (Figure 6—figure supplement 3A-B). There were 669 events that were common to both genotypes tested by Sun et al., and of these, 556 were also identified in our own RNA-seq analysis as alternatively spliced in *fio1-3* mutants (Figure 6—figure supplement 3C).

Sun et al., also performed one replicate each of nanopore DRS for Col-0, *fio1-1* and *fio1-5.* We could identify splicing changes in *AtSAR1* and *WNK1* using these data as well (Figure 6—figure supplement 4), confirming the findings we reported in our submitted manuscript.

Overall, we find that data from recently published studies support the impact of FIO1 on splicing. We have revised our manuscript to address this issue in a new Results sub-section entitled “Reanalysis of RNA-seq data from independent *fio1* alleles confirms disruption to splicing” and in the Discussion subsection entitled “FIO1-dependent m^6^A modification of U6 snRNA determines splicing accuracy and efficiency”.

In addition, FLC mRNA can be m6A methylated. The authors appear to suggest the effect is secondary. More analysis and explanation are required.

There seems to be a slight misunderstanding here. We suggest in our manuscript that because there is no change in splicing of *FLC* in the *fio1* mutant, the impact of loss of FIO1 on *FLC* expression levels occurs indirect of the methylation of U6 snRNA by FIO1 (see the context on page 19 of our submitted manuscript). We do not make any statements about the methylation of *FLC* directly, either by FIO1 or the MTA-writer complex. To avoid potential confusion to the reader, we have removed this phrase from the revised manuscript.

For instance, the authors could measure m6A level on FLC in fio1 mutants and mta mutants, and compare them with that of wild tpye.

It is true that two recent studies (Sun et al., 2022; Wang et al., 2022) reported FIO1-dependent methylation of *FLC* mRNA, but neither study examined whether the reported *FLC* modification depended upon the MTA-writer complex and neither tested the direct functional impact of the modification by mutation of the modified base(s). We have previously published in *eLife* that there is a major reduction in detectable expression of sense and antisense RNAs at the *FLC* locus in mutants defective in the MTA-writer complex component, VIRILIZER (Parker et al., 2020). In the original submitted version of this manuscript, we report a similar finding in *fio1-3* (Figure 3—figure supplement 2H-J). Consequently, we are unable to confirm the presence of m^6^A on *FLC* mRNA because the lack of *FLC* read coverage in mutants defective in either FIO1 or MTA-writer complex components means that we do not have the statistical power to do so using a comparative approach with nanopore DRS. In our view, the low levels of *FLC* transcripts also pose a technical barrier to the meaningful measurement of *FLC* m^6^A using m^6^A-IP (MeRIP) in *fio1* and *mta* mutant backgrounds. Both Wang et al., 2022 and Sun et al., 2022 (who also reported reduced *FLC* expression in *fio1* mutants) used m^6^A-IP-qPCR to address this question, but while Wang et al. reported that they used an m^6^A modified oligo spike-in to control for m^6^A levels, neither study reported how they controlled for the major difference in *FLC* input levels between wild type and *fio1* mutants. Sun et al. were not able to confirm the methylation of *FLC* using nanopore DRS. Previous studies do not agree on which methylase complex targets *FLC*, with Wang et al. & Sun et al. suggesting that FIO1 is responsible (Sun et al., 2022; Wang et al., 2022), and a third study suggesting that the MTA-complex is responsible (Xu et al., 2021). Overall, our analysis of mRNA methylation in *fip37-4* and *fio1* backgrounds indicates that FIO1-dependent mRNA methylation is rare or absent; we do not have evidence to indicate that *FLC* is an exception.

2. The authors used Nanopore m6A sequencing to map m6A in mRNA from wt and fio1 mutant strains. We would suggest either RIP-seq or mass spectrometry measurement to confirm the loss of fio1 leads to limited mRNA m6A changes.

We are grateful for this suggestion. We have added Liquid Chromatography Tandem Mass Spectrometry (LC:MS/MS) data to the revised manuscript that shows *fio1-1* mutants have 6.7% less m^6^A in poly(A)+ purified RNA than WT Col-0. This relatively modest reduction in m^6^A levels is consistent with recently published LC:MS/MS data of Arabidopsis *fio1* mutants from Wang et al. 2022. These data are presented as Figure 2A in the revised manuscript, and we have revised the Methods, Results and Discussion sections accordingly.

3. p 3 and Figure 1: "Figure 1. Schematic outline of sequential steps of pre-mRNA splicing." I don't see why this is a main figure. It's the type of schematic that could be found in any splicing review. This seems like a Supplemental figure, at most.

We are grateful for this suggestion and in response we have now removed this figure from the revised manuscript.

Instead, a schematic of their screen, described in the first paragraph of the Results, including some pictures of WT and early flowering Arabidopsis mutants obtained from the screen, would be more appropriate (and is needed).

We are grateful for this suggestion. We have included a new schematic of our mutant screen as revised Figure 1A.

We include as Figure 1 —figure supplement 1B a photograph of the actual F2 segregating population used to map *fio1-4*, including phenotypic early flowering plants and aphenotypic sisters. For additional clarity, we have incorporated schematic images of the early flowering individuals in Figure 1A, which outlines how the mutant screen was performed. A photograph of the early flowering phenotype of the *fio1* alleles used in this study compared to WT is presented as Figure 1F in the revised manuscript.

The authors should also show the RT-PCR data supporting enhanced MAF2 intron 3 retention at 16{degree sign}C.

We provide here Figure 1—figure supplement 1A , the raw RT-PCR data from the second step of the two-step mutant screen that includes the EMS129 line (later referred to as *fio1-4*) where plants were cultivated at 16°C. In addition, we have included this schematically in Figure 1A outlining how the mutant screen was performed.

Overall, the authors provide few details about this screen, including how many EMS mutants were screened.

We agree that our description of the screen that identified the *fio1-4* allele required more detail. We have now prepared a re-analysis of the data from our bulk segregant sequencing and updated the Results, Methods, and figures accordingly. 15,000 EMS M2 seeds were screened using a two-step procedure. The earliest flowering 100 EMS lines at 16°C were selected for splicing analysis by RT-PCR. This approach identified two lines that had increased retention of *MAF2* intron 3. We focussed this study on EMS129. The causal mutation for the early flowering phenotype of EMS129 was mapped by backcrossing to the M0 parental line to generate a segregating population. Early flowering plants and aphenotypic sisters were grouped into separate pools for genomic DNA sequencing. We identified G to A transitions caused by EMS mutations using a SNP-calling pipeline, which is now described in greater detail in the revised Methods section. 19 SNPs in a 2.6Mb region on Chromosome 2 were identified as significantly associated with the early flowering phenotype (G test FDR < 0.05, allele fraction 100%). These SNPs and the associated allele fractions are shown in revised Figure 1B in the revised manuscript. No other associations were found on other chromosomes. Of these SNPs, only one was predicted to have a major effect on gene expression: a G to A transition at the +1 position of the 5’SS in the second intron of *FIONA1*.

How did the authors identify mutants that alter MAF1 intron 3 splicing, when the difference in intron 3 retention (as shown in Figure 4C) is so subtle?

This is explained by the fact that we carried out a two-step mutant screen. MAF2 represses flowering at low ambient temperature. We wanted to identify factors required for the efficient splicing of *MAF2* intron 3 at low ambient temperature. We reasoned that if *MAF2* intron 3 was spliced less efficiently, the level of MAF2 flowering repressor activity would be reduced and hence plants would flower earlier. In step 1, we screened 15,000 EMS mutants for early flowering individuals grown at 16°C. In step 2, we re-screened the earliest flowering 100 mutant individuals for changes in *MAF2* intron 3 splicing by RT-PCR. Although the difference in *MAF2* intron 3 processing is indeed relatively modest, we report multiple changes in the splicing and expression of other genes controlling flowering time in *fio1* mutants in our manuscript. Consistent with this, *fio1* mutants have a very clear early flowering phenotype, and indeed, it was through an early flowering mutant screen that Hong Gil Nam’s lab first isolated the *fio1-1* mutant allele (Kim et al., 2008). In the second step of the screen, we used RT-PCR to analyse *MAF2* intron 3 splicing in RNA purified from these 100 early flowering mutants. Two of these 100 early flowering individuals showed increased levels of *MAF2* intron 3 retention, one of which is the *fio1-4* allele that we describe here. Although the difference in *MAF2* intron 3 splicing is relatively small, it is clearly identifiable by RT-PCR (see Figure 1—figure supplement 1A ) and confirmed by our RT-qPCR analysis of other alleles (Figure 1D) and Illumina RNA-seq of another allele (see Figure 3 C and D in the revised manuscript).

We believe that the Reviewer’s suggestion to include a schematic on the mutant screen helps to clarify how this two-step screen could identify mutants with relatively small differences in the splicing of *MAF2* intron 3 and clarifies this point for the reader. We have also incorporated the phrase “two-step mutant screen" in different sections of the revised manuscript to reinforce this point.

4. p 6. "Crosses between EMS 129 and either fio1-1, or a Transfer-DNA insertion line disrupting AT2G21070 (SALK_084201; fio1-3) confirmed allelism." No data for this are shown; data supporting these conclusions should be included.

We apologise for this oversight and the manuscript has now been revised to provide these data. 100% of EMS129 lines crossed to the *fio1-1* mutant flowered early (35 plants), compared to 0% of EMS129 lines crossed to Col-0 (15 plants), confirming allelism. These phenotypes were scored visually at the time of allelism testing and leaf count data were not recorded. Our U6 snRNA m^6^A analysis reveals that all the alleles studied here (*fio1-1*, *fio1-3* and *fio1-4*) are defective in U6 snRNA m^6^A, independently confirming the allelism test of flowering phenotype, and these data are reported later in the manuscript text (Figure 2D of our revised manuscript). Our global RNA analyses are based on sequencing of both *fio1-1* and *fio1-3* alleles, which independently validate our conclusions on the impact of FIO1 on splicing. Several other *fio1* alleles disrupting AT2G21070 that flower early and lack U6 snRNA m^6^A modification have been reported this year (Sun et al., 2022; Wang et al., 2022; Xu et al., 2022). As we show in response to comment 1 in this document, the splicing phenotypes are shared by these different alleles as revealed by our reanalysis of the data published in these studies. Consequently, our analysis of *fio1-1, fio1-2, fio1-3, fio1-4* and *fio1-5* support the conclusion that all of these alleles exhibit specific, overlapping splicing defects. The text in the Results section has been revised accordingly.

5. Many of the figures are difficult to interpret. For example, Figure 3 legend gives some details of the bioinformatics, but does not say anything about what the actual experiments were. The text is overly-focused on bioinformatics without an adequate description of the experiments that were performed. Please provide information detailing the actual experiments; without this information, these parts of the manuscript are both uninformative and impossible to evaluate.

We agree that our Figure Legends (and the Results text referring to them) were inadequately described. We have systematically reviewed and revised all figure legends and how they are referred to in the text to clarify their content and explain the analyses of large datasets in a more accessible way.

6. Figure 4C. The change in MAF2 intron 3 retention is seemingly tiny. What exactly is the deltaPsi for this? Can this be confirmed by RT-PCR or RT-qPCR? This highlights an issue raised above: how did the authors identify mutants in the screen by RT-PCR that altered MAF2 intron 3 retention if the change is this subtle?

The ΔPSI measured in our Illumina RNA-seq data is 0.066 representing a 6.6% increase in intron retention in the *fio1-3* mutant across all temperature conditions (FDR = 2.7×10^–3^). The increase in *MAF2* intron 3 retention in a *fio1-4* mutant was initially identified using RT-PCR (during the mutant screen, Figure 1—figure supplement 1A ) and splicing differences were confirmed by RT-qPCR in *fio1-1* and *fio1-3* alleles (Figure 1D of the revised manuscript). Our detailed explanation as to how the mutant was identified is given in response to the same question raised in point 3.

7. I found the description of the SS sequence enrichment incredibly long (7 pages of text) and overwhelmingly descriptive without a view of what the conclusions would be.

We agree that this section of the Results is long and detailed, but we are keen to retain some of this format because it is central to the advances that we make and it provides insight that other studies on mutants defective in FIO1/METTL16 have missed. We have edited each Results section and removed some of the numeric terms to improve readability in the revised manuscript. We have cut 347 words from the SS enrichment section.

We deliberately chose to submit this work to *eLife* because of its objectives around transparency, open access and transforming academic publishing. It is difficult to interpret “overwhelmingly descriptive” in the context of the series of novel findings that we report in this section. Eve Marder, who was a Deputy Editor of *eLife* 2015-2019, published an essay in *eLife* remarking how in manuscript reviews, terms like “mechanism” and “descriptive” are often used as pejoratives without meaning (Marder, 2020). Consequently, it is disappointing to see such expressions reach through to this stage of an *eLife* review.

Sequence motifs in the context of otherwise disparate intron and exon sequences can be suggestive of a mechanism; but, in the end, confirmation of these sequence effects requires experimental testing within a constant gene sequence. This is necessary to make any solid conclusion based on rigorous science. These are critical experiments that are missing in this manuscript.

Our manuscript does report an experimental test within a constant gene sequence. Our *fio1* mutant alleles lack m^6^A modification of U6 snRNA but the nucleotide sequence of splice sites that are used in WT or *fio1* mutant alleles are unchanged between these genetic backgrounds genome-wide. We measured splicing that occurred in vivo in otherwise controlled conditions, in multiple cell types of whole organisms exposed to different environmental temperature regimes. High throughput in vivo analyses like ours show that effects which are identifiable in individual introns, like *MAF2* intron 3 or *AtSAR1* intron 21, are explainable by simple changes in nucleotide preference which generalise across thousands of different contexts. Similar perturbations of the U5 snRNA have previously been reported: as we described on p20 of our submitted manuscript “…defective splicing of specific introns in *Schizosaccharomyces pombe mtl16* mutant strains can be experimentally rescued by expression of mutated U5 snRNAs designed to strengthen U5 loop 1 interactions with the upstream exon (Ishigami et al., 2021)”. Importantly, these “genetic complementation” experiments are consistent with the statements that we make with respect to the global distinction of two 5’SS classes. Therefore, there is already orthogonal experimental support for our model.

An alternative interpretation of the reviewer’s comment (“experimental testing within a constant gene sequence”) could be the use of single transgene reporters with base substitutions e.g. at the +4 position of the 5’SS. While we recognise the value of different experimental approaches, our view is that our unbiased global approach is more “rigorous”, because in reality it is not so simple to derive broad conclusions from testing individual reporter genes. This is because splicing outcomes are the result of complex combinatorial inputs, and so the impact of changes in an example reporter gene may not be generalisable to other sequence contexts. Several recent publications illustrate this point:

First, we cited the high throughput study from Adrian Krainer’s lab (Wong et al., 2018), which used a massively parallel splicing assay with minigene constructs to determine the efficiency of all possible 9 nt 5’SS sequences (32,678 sequences) in three different intron contexts. A major conclusion of this comprehensive analysis is that 5’SS preferences were different depending on the broader context of the minigene.

Second, a recent study from Stirling Churchman’s lab (Choquet et al., 2022) used nanopore direct RNA sequencing analysis of human chromatin-associated poly (A)+ RNA to reveal that post-transcriptional splicing of multiple introns within the same transcript follows a defined or preferred order. This illustrates that testing sequences in one intron can be confounded by the combinatorial impact that other introns in the same transcript have on the splicing of the intron under experimental examination (this was also a conclusion of the study from the Krainer lab).

Third, a recent study from Jean Beggs’ lab (Aslanzadeh and Beggs, 2020) shows that the efficiency of splicing is influenced by the rate of RNA Polymerase II (PolII) elongation. Jean Beggs’ study is particularly pertinent here because she looked at *Saccharomyces cerevisiae* ribosomal protein genes which possess a C, rather than the consensus U at the 5’SS+4 position. Compared to WT, in lines with a slow elongating PolII mutant, non-consensus 5’SS introns were spliced more efficiently, but in lines with a fast-elongating PolII mutant, such introns were spliced less efficiently. Consequently, interpreting the impact of individual positions in 5’SSs could be confounded not only by cis-elements in the test RNA but by the promoter used to drive the expression of otherwise constant gene sequences.

An expectation from the *eLife* review process is that “extra experiments, analyses, or data collection are only requested if they are essential and can be reasonably completed within about two months”. Because the critical experiments are not specified, it is unclear whether they could be completed in Arabidopsis in such a time frame. For example, the production of stable transgenic Arabidopsis reporter lines takes several months.

8. The model seems to be that U5 and U6 RNAs contribute to a platform for recognition of 5'SSs: a 5'SS must interact sufficiently with this platform in order to make it through splicing;

A key tenet of our study is indeed that U5 and U6 snRNAs act together as a platform in which stronger interactions with either U5 or U6 can compensate for weaker interactions with the other. We demonstrate this through global RNA-Seq analyses using *fio1* mutants that alter the sequence preference of U6 snRNA. We show that the use of 5’SSs that have weaker U6 snRNA interactions in *fio1* mutants is less affected when these 5’SSs have stronger U5 interacting sequences.

Stronger interaction as some positions can compensate for weaker interactions at other positions.

This is not completely correct. Rather than finding that changes in any 5’SS position can compensate for weaker interactions resulting from loss of U6 snRNA m^6^A, we find specific changes (namely, a change in nucleotide preference at the +4 position and a stronger U5 interacting potential at the –2 and –1 positions) define the altered 5’SS motif for sites used in the *fio1* mutant.

I don't think that there's anything new here in terms of mechanism.

Interpretation of this statement is difficult because it is unclear what level of understanding constitutes new “mechanistic” insight (Marder, 2020). Our study expands upon previous work by extending the model of U5/U6 compensation from intron retention events to more complex forms of alternative splicing with two or more competing 5’SSs. This includes not only alternative 5’SSs, but exons with increased and decreased skipping. In each case we show that the model of compensatory U5 and U6 interactions can be used to explain splicing changes. Furthermore, we show that increases in upstream and downstream alternative 3’SS preference can also be explained by a weakening or strengthening of the 5’SS:U6 interaction, respectively. Finally, we provide a detailed analysis of available cryo-EM data to contextualise these findings. We have restructured the Discussion section of the revised manuscript to make these points more clearly.

m6A in U6 contributes to stability; this was already postulated in the S. pombe system (Ishigami et al. 2021).

It is true that Ishigami et al. postulated that m^6^A contributes to the stability of U6 – 5’SS interactions. However, our cryo-EM analysis provides a more detailed interpretation of how this might occur and goes further to incorporate our novel finding on U6 snRNA affecting 3’SS usage. We report that U6_A43_ faces 5’SS A_+4_ in a trans Hoogsteen sugar edge interaction that could stabilise the U6/5’SS helix in B complex by capping. We show that this stabilisation may be particularly important in B but not B^act^ complex because in B^act^ the U6/5’SShelix is now capped by 5’SSA_+3_ interacting with U6 snRNA G44. We then report that this helix capping property of A43 becomes important again in C* complex consistent with our RNA-Seq data that reveal an impact of U6 snRNA m^6^A modification on 3’SS usage. None of these insights are included in the *S. pombe* study reported by Ishigami et al., or any other study to date.

The agreement between our manuscript and the *S. pombe* study by Ishigami et al. is an important strength of our manuscript because together we establish that the conserved role of METTL16/FIONA1 is to methylate U6 snRNA and that this has demonstrable and consistent impacts on pre-mRNA splicing. This is important because recent studies have suggested an alternative role for FIO1 in methylating mRNA that is not supported by our data.

Is this biologically important? Does FIO1 expression change developmentally? or, is it a constitutive feature of U6-5'SS interaction? These questions are not addressed or even raised in this manuscript.

Questions of whether the regulation of *FIONA1* and/or U5/U6 compensation could be used to control alternative splicing were addressed in the submitted manuscript. The final Results section on page 18 of the submitted manuscript addresses this: we show that pairs of alternative 5’SSs in Arabidopsis, humans, and other species tend to have opposing U5 and U6 interaction potential, which could be used to modulate alternative splicing. In addition, we dedicated the entire final section of the Discussion to the question of regulation – see page 24 headed “A regulatory or adaptive role for U6 snRNA modification?”

Analysis of publicly available Arabidopsis community resources (eg.: bar.utoronto.ca) suggests that *FIO1* mRNA is constitutively expressed (Waese et al., 2017). This does not preclude a condition-specific impact of FIO1 on splicing. Furthermore, constitutive expression of *FIO1* mRNA does not preclude regulation of the protein’s activity or targeting. Consequently, it remains an open question as to whether m^6^A levels of U6 snRNAs vary.

U6 snRNA methylation may be a constitutive feature of the U6:5’SS interaction. Nevertheless, its loss has very specific phenotypic impacts, for example on flowering time. Consequently, it is biologically important, regardless of whether it is constitutive or not. Likewise, an example from the splicing field is that mutation of the Brr2 constitutive splicing factor leads to a very specific human disease, Retinitis Pigmentosa (Zhang et al., 2009). Furthermore, whilst loss of m^6^A from U6 aided in the identification of opposing interaction potential of alternative 5’SS pairs, it does not mean that this is the only mechanism by which U5 and U6 interactions could be regulated.

As we discussed in our submitted manuscript, Arabidopsis flowering time GWAS analyses point to a functional impact of natural variation in the sequence of *FIO1* genes (Price et al., 2020; Sasaki et al., 2015). We have started a new research programme to understand the impact of this sequence variation, but it is outside the scope of the present study.

We have restructured the Discussion to make these points clearer in our revised manuscript.

9. p 18. The authors state that "FIO1 buffers spicing fidelity" and "that FIO1 function calibrates the temperature range over which MAF2 alternative slicing occurs", but I don't see any data that directly support these assertions. These seem like extrapolations of the current data that require experimental tests to support them.

We agree that these statements need clarifying and we have removed these phrases to make the reporting of our data clearer:

In the course of re-writing the manuscript we have removed the Discussion sub-section entitled “FIO1 buffers splicing fidelity”. A more straightforward way to describe this finding would be “Loss of FIO1 introduces temperature sensitivity to some splicing events”. The data that supported this statement was described in the Results section on page 10 and Figure 3 —figure supplement 1C of the original submitted manuscript (now Figure 3 —figure supplement 1C in the revised manuscript). The data shown in Figure 5D of the submitted manuscript (now Figure 4D in the revised manuscript) is also consistent with this statement. These data are retained in the revised manuscript.

Another way to state “that FIO1 function calibrates the temperature range over which MAF2 alternative splicing occurs” could be the more straightforward “In the absence of FIO1, the temperature range over which variation in MAF2 intron 3 retention occurs is shifted”. The data that supports this statement is shown in Figure 3D of the revised manuscript. Nevertheless, for clarity on this issue, we have removed this phrase from the revised manuscript.

10. The authors refer to the "cooperative roles for U5 and U6 snRNA in splice site selection" in the abstract. I don't see any evidence for interactions with U5 in this manuscript,

Our global analyses document 1000s of splicing events sensitive to FIO1 mutation that can be distinguished by the base identities of 5’SS-U5 snRNA interacting positions. The recognition of –1 and –2 exon positions at 5’SSs by U5 snRNA has been established over a period of approximately 40 years, using techniques ranging from genetic analysis with yeast to cross-linking studies, high-throughput sequencing experiments and direct visualisation using cryo-EM (Wilkinson et al., 2020). It is well established that the –2 and –1 positions interact with the U5 snRNA, and that an AG motif at the –2 to –1 positions of the 5’SS leads to more efficient splicing (Wong et al., 2018). It is similarly uncontroversial that the +3 to +5 positions are important for the interaction with U6 snRNA, and that an RAG motif at the +3 to +5 positions leads to more efficient splicing (Wong et al., 2018). Our analyses of all genomic 5’SS sequences (in Arabidopsis and a range of metazoans) show that when 5’SSs have an AG motif at the –2 to –1 positions of the 5’SS, the requirement for an RAG motif at the +3 to +5 positions is relaxed. This finding is presented in Figure 7C of our manuscript. Furthermore, our RNAseq data shows that the presence of an AG motif at the –2 to –1 positions is able to compensate for a weakened interaction with the U6 snRNA caused by loss of U6 snRNA methylation in *fio1* mutants. We showed this using the effect size of alternative splicing changes classified by motif at the –2 to –1 positions, which we presented in Figure 4H of the manuscript. The –2 to –1 positions interact with the U1 and U5 snRNAs, and the interaction with U1 snRNA occurs prior to handover to U5/U6 and hence should be unaffected by loss of U6 snRNA methylation. Therefore, our findings can be explained by stronger U5 snRNA interactions compensating for weakened U6 interactions, supporting a model of cooperative and compensatory roles for U5 and U6 snRNAs in splice site selection. A similar result was also recently reported in *S. pombe* by Ishigami et al. using RNA-seq data and validated using genetic suppression experiments with modified U5 loop 1 sequences (Ishigami et al., 2021).

and the proposed role of U6 is not tested experimentally. These two deficits are the most disappointing aspects of this work.

This phrasing is somewhat ambiguous because our study is focused on the role of m^6^A modification of U6 snRNA. The role of the U6 snRNA in 5’SS recognition is well established (Wilkinson et al., 2020). The methylation of the central adenosine of the ACAGA box by METTL16 is well established in other eukaryotes (Epstein et al., 1980; Kiss et al., 1987), and we have used m6A-IP qPCR experiments to demonstrate that this methylation is conserved in the Arabidopsis U6 snRNA and directed by the METTL16 homolog FIONA1. We have performed RNA-seq experiments with extremely high levels of depth and replication to dissect the consequences of the loss of U6 snRNA methylation on splicing, and provide multiple in-depth analyses to show that in all cases these consequences can be explained by changes in sequence preference at the +4 position of the 5’SS. We provide multiple lines of evidence from our RNA-seq and 5’SS sequence analyses to show that there are compensatory interactions between U5 and U6 snRNAs in their role in 5’SS recognition. Consequently, it is unclear which aspect of the “proposed role of U6” we have failed to test.

We have used a detailed analysis of the arrangement of the U5 and U6 snRNAs in multiple cryo-EM datasets to interpret the findings of our RNA-seq analyses. A potential follow-up experiment is cryoEM analysis of human spliceosomes prepared with U6 snRNA lacking m^6^A modification (in vitro splicing assays and cryo-EM using Arabidopsis spliceosomes is currently not possible). This would be a major undertaking, however, and these experiments are not feasible within the timeframe for paper revisions recommended by *eLife*.

An expectation from the *eLife* review process is that “extra experiments, analyses, or data collection are only requested if they are essential and can be reasonably completed within about two months”. Because the experiments raised here are not specified, it is unclear whether they could be completed in Arabidopsis in such a timeframe.

Additional comments:1. The main text goes on for 31 pages, >12,000 words (not including Figure Legends or References). While there are no strict limits on the length of Research Articles, the eLife Author Guide suggests to "try not to exceed 5,000 words in the main text". The text could be shortened significantly; conciseness would greatly benefit the readability of the manuscript.The Discussion, at >7 pages, is excessively long and does not critically evaluate the data or conclusion presented in this manuscript.

The *eLife* guidelines state: “There is no maximum length for Research Articles, but we suggest that authors try not to exceed 5,000 words in the main text, excluding the Materials and methods, References, and Figure legends.” The word count of the main text that we submitted was 8,952. We have edited each section in the revised manuscript to reduce the word count and improve the readability. In particular, we have reduced the length of the discussion from 2809 words to 2050 words and shortened the Results sections reporting analysis of our RNA-seq experiment from 3227 words to 2880 words. Due to the additional information requested by the reviewers in other areas of the Results section, the overall length of the results has not materially changed: going from 4795 words to 4906 words. The overall length of the main text is now 8,208.

Our manuscript is relatively long because of the range of subjects addressed:

Temperature-sensitive control of floweringFIO1-dependent modification of U6 snRNA and poly(A)+ mRNAExplanation of the early flowering phenotype of *fio1* mutantsImpacts of U6 m^6^A on splicingDefinition of two classes of 5’ SS in diverse eukaryotesInterpretation of cryo-EM to explain the role of U6 m^6^A in 5’SS and 3’SS usageClarification of recently published studies that mistakenly report no effect of loss of U6snRNA m^6^A modification on splicing

And the different experimental approaches used:

Arabidopsis mutant screening, mapping and phenotypingMolecular analysis of RNA (RT-PCR, Immunoprecipitation of RNP complexes)Illumina RNA-seq, nanopore direct RNA-sequencing with splicing and RNA modification analysisCryo-EM

This is a rare combination of subjects and approaches that requires attention in the Introduction and Discussion sections. We see the diversity and orthogonal nature of the approaches that we have used as a strength of the manuscript. As a result, we highlight larger implications of our study in the Discussion. We do not agree that the Discussion fails to “critically evaluate the data or conclusion”.

2. p 7 and Figure 2B. What is the experimental basis of Figure 2B? The authors do not state the experimental data: is it RNA-seq or RT-PCR? In either case, some examples of the primary data for this should be shown.

We apologise for this oversight. The data derive from RT-qPCR analysis. We have revised the labelling of the Figure and associated Figure Legend to make this clear. The raw source data for this figure was shared at the time of submission and is available associated with the bioRxiv pre-print. In the current version of the manuscript, it is Figure 1 source data 3.

3. p 6, second paragraph of the Results: "We used bulked segregant analysis and the software tool artMap (Javorka et al., 2019) to identify the causative mutation…". The authors should explain this more explicitly, for non-specialist readers. There are a number of ways in which this could have been done, but the authors do not specify what exactly was the data that led to the determination of the mutation.

As described above, we have updated the Methods and Results sections to explain the screen and the processing of the bulk segregant analysis data more clearly.

4. Figure 4B. This is a comparison of fio1-3 with Col-0. I would like to see volcano plots for all the five splicing events with log2FC on the x-axis and enrichment/significance on the y axis. These bar charts with percentages are too confusing to depict what the authors are actually trying to convey and do not depict the spread of the primary data.

We agree that the bar chart is a high-level visualisation of the data which does not allow readers to interpret effect sizes or significance levels. We have taken the advice of the reviewer and created volcano plots for the four main alternative splicing event types that are focused on in the paper. These volcano plots also show that the most significant effects of loss of FIO1 are changes in alternative 5’SS usage and increases in intron retention. We have included these volcano plots as Figure 3—figure supplement 1B in the revised manuscript.

5. Figure 4E shows "absence of change" in MAF2 intron 5, but this intron is clearly efficiently spliced, so the comparison to inefficiently-spliced intron 3 is not particularly meaningful. A more meaningful comparison would be to a different intron that is also inefficiently spliced.

This comparison was intended to exemplify that FIO1 is not required for the splicing of all introns, by comparison to another intron in the same gene (*MAF2*). However, we agree that this is not a particularly useful control since efficiently spliced introns may not be as sensitive to splicing perturbations. Consequently, we have removed this figure.

6. Figure 5. Seq logos are shown for 5'SSs that are sensitive to or that are used better in fio1-1 mutants. These can only be meaningful if they are compared to the seq logo of the overall 5'SS in A.t., which is not shown or discussed.

We have now added the sequence logo of all annotated 5’SSs in the condition-specific assembly as Figure 4 —figure supplement 1B in the revised manuscript. Sequence logos for 5’SSs can now be compared to this overall logo. However, we also intend for comparisons to be drawn directly between 5’SSs that are used more in the *fio1-3* mutant and their cognate 5’SSs that are used less. As a control to show that the strong difference in sequence preference caused by loss of U6 snRNA methylation is unusual, we included sequence logos and heatmaps for alternative 5’SSs that are used more at high temperatures and cognate sites that are preferred at low temperatures (compare Figure 4 —figure supplement 1C-D in the revised manuscript). These two sets of matched 5’SSs show no overall difference in sequence preference as reported on page 11 of our submitted manuscript.

7. Figure 5. Why only show out to pos -2 in the exon? More positions than that can pair with U5.

It is true that more positions can base pair with U5 snRNA loop 1. In the original submitted version of our manuscript, we tried to consider this carefully with regard to both the sequence logos and the heatmaps of U5 and U6 interacting sequences. Although the -3 position of the 5’SS can pair with U5, we found that the information content of the -3 position in annotated Arabidopsis 5’SSs is low with only 0.10 bits of entropy compared to 0.49 and 0.85 bits for the -2 and -1 positions respectively. The entropy of the -4 position is even lower (0.04 bits) suggesting almost no sequence preference at this position. We found that when aggregating data into U5 and U6 heatmaps, the -3 position did not contain important information, making the heatmaps more complex to interpret (see Author response image 3 for an example). Hence, we chose to exclude the -3 position from heatmaps and sequence logos in our submitted manuscript. However, we recognise that it is standard practice to display 5’SS sequence logos from the -3 position to +5 position, and this does not overly complicate sequence logo interpretation. We have therefore redrawn sequence logo figures to include the -3 position of the 5’SS in Figures throughout the revised manuscript.

**Author response image 3. sa2fig3:** Heatmaps showing the distribution of U5 snRNA and U6 snRNA interacting sequence classes for 5’SSs of introns which have increased (left) and decreased (right) retention in *fio1-3*. U5 classes are based upon the distance of the —3 to —1 positions of the 5’SS from the consensus motif AAG. U6 classes are based upon the distance of the +3 to +5 positions of the 5’SS from the consensus motif RAG.

8. Some of the fonts in the figures are so small as to be unreadable or nearly unreadable, e.g. Figure 4-Figure sup 1 and Figure 5-Figure sup 1.

The figures are scaled to the size of the page in the submitted PDF, but we also submitted full resolution figures matching *eLife* recommendations (>10cm, >300dpi) with uniform font sizes across all figures (8pt). If our manuscript is accepted for publication, we will consult the production editor for direction on how to appropriately resolve this issue.

9. ALKBH10B is a demethylase that affects early flower phenotype. Does this protein also mediate the demethylation of U6 m6A?

We are grateful for this interesting suggestion, and we have incorporated it into the revised Discussion section. It is true that *alkbh10b* mutants have been reported to be late flowering (Duan et al., 2017). It is plausible that ALKBH10B could demethylate U6 snRNA, since a related human protein, FTO, has been shown to remove TMG-cap adjacent m6Am modifications from snRNAs (Mauer et al., 2019). To experimentally test this, however, would require an *alkbh10b* mutant, which we currently do not have. This line of research is therefore not feasible in a 2-month period but it will be interesting to examine in the future.

References

Aslanzadeh V, Beggs JD. 2020. Revisiting the window of opportunity for cotranscriptional splicing in budding yeast. *RNA* 26:1081–1085.

Choquet K, Koenigs A, Dülk S-L, Smalec BM, Rouskin S, Churchman LS. 2022. Pre-mRNA splicing order is predetermined and maintains splicing fidelity across multi-intronic transcripts. *bioRxiv*. doi:10.1101/2022.08.12.503515

Duan H-C, Wei L-H, Zhang C, Wang Y, Chen L, Lu Z, Chen PR, He C, Jia G. 2017. ALKBH10B Is an RNA N 6-Methyladenosine Demethylase Affecting Arabidopsis Floral Transition. *Plant Cell* 29:2995– 3011.

Epstein P, Reddy R, Henning D, Busch H. 1980. The nucleotide sequence of nuclear U6 (4.7 S) RNA. *J Biol Chem* 255:8901–8906.

Ishigami Y, Ohira T, Isokawa Y, Suzuki Y, Suzuki T. 2021. A single m6A modification in U6 snRNA diversifies exon sequence at the 5’ splice site. *Nat Commun* 12:3244.

Kim J, Kim Y, Yeom M, Kim J-H, Nam HG. 2008. FIONA1 is essential for regulating period length in the Arabidopsis circadian clock. *Plant Cell* 20:307–319.

Kiss T, Antal M, Solymosy F. 1987. Plant small nuclear RNAs. II. U6 RNA and a 4.5SI-like RNA are present in plant nuclei. *Nucleic Acids Res* 15:543–560.

Marder E. 2020. Words without meaning. *ELife* 9. doi:10.7554/*eLife*.54867

Mauer J, Sindelar M, Despic V, Guez T, Hawley BR, Vasseur J-J, Rentmeister A, Gross SS, Pellizzoni L, Debart F, Goodarzi H, Jaffrey SR. 2019. FTO controls reversible m6Am RNA methylation during snRNA biogenesis. *Nat Chem Biol* 15:340–347.

Parker MT, Knop K, Barton GJ, Simpson GG. 2021. 2passtools: two-pass alignment using machinelearning-filtered splice junctions increases the accuracy of intron detection in long-read RNA sequencing. *Genome Biol* 22:72.

Parker MT, Knop K, Sherwood AV, Schurch NJ, Mackinnon K, Gould PD, Hall AJ, Barton GJ, Simpson GG. 2020. Nanopore direct RNA sequencing maps the complexity of Arabidopsis mRNA processing and m6A modification. *ELife* 9. doi:10.7554/*eLife*.49658

Price N, Lopez L, Platts AE, Lasky JR. 2020. In the presence of population structure: From genomics to candidate genes underlying local adaptation. *Ecol Evol* 10:1889–1904.

Sasaki E, Zhang P, Atwell S, Meng D, Nordborg M. 2015. ‘Missing’ G x E Variation Controls Flowering Time in *Arabidopsis thaliana*. *PLoS Genet* 11:e1005597.

Shen S, Park JW, Lu Z-X, Lin L, Henry MD, Wu YN, Zhou Q, Xing Y. 2014. rMATS: robust and flexible detection of differential alternative splicing from replicate RNA-Seq data. *Proc Natl Acad Sci U S A* 111:E5593-601.

Sun B, Bhati KK, Song P, Edwards A, Petri L, Kruusvee V, Blaakmeer A, Dolde U, Rodrigues V, Straub D, Yang J, Jia G, Wenkel S. 2022. FIONA1-mediated methylation of the 3’UTR of FLC affects FLC transcript levels and flowering in Arabidopsis. *PLoS Genet* 18:e1010386.

Tang AD, Soulette CM, van Baren MJ, Hart K, Hrabeta-Robinson E, Wu CJ, Brooks AN. 2020. Full-length transcript characterization of SF3B1 mutation in chronic lymphocytic leukemia reveals downregulation of retained introns. *Nat Commun* 11:1438.

Trincado JL, Entizne JC, Hysenaj G, Singh B, Skalic M, Elliott DJ, Eyras E. 2018. SUPPA2: fast, accurate, and uncertainty-aware differential splicing analysis across multiple conditions. *Genome Biol* 19:40.

Waese J, Fan J, Pasha A, Yu H, Fucile G, Shi R, Cumming M, Kelley LA, Sternberg MJ, Krishnakumar V, Ferlanti E, Miller J, Town C, Stuerzlinger W, Provart NJ. 2017. EPlant: Visualizing and exploring multiple levels of data for hypothesis generation in plant biology. *Plant Cell* 29:1806–1821.

Wang C, Yang J, Song P, Zhang W, Lu Q, Yu Q, Jia G. 2022. FIONA1 is an RNA N6-methyladenosine methyltransferase affecting Arabidopsis photomorphogenesis and flowering. *Genome Biol* 23:40.

Wilkinson ME, Charenton C, Nagai K. 2020. RNA splicing by the spliceosome. *Annu Rev Biochem* 89:359–388.

Wong MS, Kinney JB, Krainer AR. 2018. Quantitative Activity Profile and Context Dependence of All Human 5’ Splice Sites. *Mol Cell* 71:1012-1026.e3.

Xu C, Wu Z, Duan H-C, Fang X, Jia G, Dean C. 2021. R-loop resolution promotes co-transcriptional chromatin silencing. *Nat Commun* 12:1790.

Xu T, Wu X, Wong CE, Fan S, Zhang Y, Zhang S, Liang Z, Yu H, Shen L. 2022. FIONA1-mediated m6 A modification regulates the floral transition in Arabidopsis. *Adv Sci (Weinh)* e2103628.

Zhang L, Xu T, Maeder C, Bud L-O, Shanks J, Nix J, Guthrie C, Pleiss JA, Zhao R. 2009. Structural evidence for consecutive Hel308-like modules in the spliceosomal ATPase Brr2. *Nat Struct Mol Biol* 16:731–739.